# Gas-Phase Fragmentation of Cyclic Oligosaccharides in Tandem Mass Spectrometry

**DOI:** 10.3390/molecules24122226

**Published:** 2019-06-14

**Authors:** Alexander O. Chizhov, Yury E. Tsvetkov, Nikolay E. Nifantiev

**Affiliations:** N. D. Zelinsky Institute of Organic Chemistry, Russian Academy of Science, Leninskii Prosp., 47, 119991 Moscow, Russia; tsvetkov@ioc.ac.ru (Y.E.T.); nen@ioc.ac.ru (N.E.N.)

**Keywords:** cyclooligosaccharides, cyclodextrins, cyclooligoglucosamines, cyclofructans, cycloglycans, cyclolaminarinoses, cyclosophoraoses, macrocycles, tandem mass spectrometry, electrospray ionization, matrix-assisted, laser-induced desorption/ionization, fragmentation, collision induced dissociation, structure elucidation

## Abstract

Modern mass spectrometry, including electrospray and MALDI, is applied for analysis and structure elucidation of carbohydrates. Cyclic oligosaccharides isolated from different sources (bacteria and plants) have been known for decades and some of them (cyclodextrins and their derivatives) are widely used in drug design, as food additives, in the construction of nanomaterials, etc. The peculiarities of the first- and second-order mass spectra of cyclic oligosaccharides (natural, synthetic and their derivatives and modifications: cyclodextrins, cycloglucans, cyclofructans, cyclooligoglucosamines, etc.) are discussed in this minireview.

## 1. Introduction

Macrocyclic oligosaccharides have been known for a long time. The most studied of them, cyclodextrins (CDs, α-CD, cyclohexamaltose, **1**; β-CD, cycloheptamaltose, **2**; γ-CD, cyclooctamaltose, **3**, Figure 1), were isolated at the end of 19th century and their cyclic structure (cycles of α-(1→4)-linked glucopyranose units) was firmly established in the middle of the last century (for the history of studies of cyclodextins and their applications, see the excellent review of Crini [1] and references therein). 

There are many other carbohydrate macrocycles, both of natural and synthetic origin [2,3]. Due to the presence of hydrophobic cavities in CD molecules, these macrocycles and their derivatives are capable to form host-guest inclusion complexes with many organic substances [4,5]. This opens possibilities for versatile applications of CDs, for example, for drug delivery of pharmaceutical formulations [5,6], construction of nanomaterials, sugar-based surfactants and food ingredients, preparations for controlled release of fragrances and aromas ([1] and references therein). CDs may be used as antidotes due to selective bonding of toxicants and disease-related metabolites (for example, see [7]: amyloid-β-peptide related to Alzheimer’s disease has strong affinity to β-CD, and hydroxypropyl-β-CD reduces its cell toxicity in model experiments). Mass spectrometry has enormous sensitivity thus allows to use this method for the analysis of trace amounts of drugs, metabolites, xenobiotics, etc. Soft ionization methods (primarily, ESI and MALDI) make possible to transfer heavy, polar molecules into the gas phase. This achievement has opened a way for complementary, mass spectrometric approaches [8] for gas-phase studies of macrocycles (including CDs) in addition to NMR, UV/Vis spectroscopy, circular dichroism, chromatography, etc. used for liquid phase studies [4]. In this review, we have attempted to present the current state of knowledge of mass spectrometry (MS) of cyclic oligosaccharides (cyclodextrins, cyclofructans, cyclo-oligosophoraoses, cyclooligolaminarioses, cyclooligoglucosamines and mixed cyclic oligosaccharides) and to discuss some problems arising from the peculiarities of their structures.

## 2. First- and Second-Order Mass Spectra of Cyclic Oligosaccharides and Their Derivatives

### 2.1. Mass Spectrometry of Cyclic Oligosaccharides—General Aspects

Oligosaccharides (OSs) are non-volatile, thermally unstable compounds so they cannot be analyzed directly by electron ionization (EI) mass spectrometry. Derivatization of oligosaccharides has limited applications, although OS derivatives were used in earlier EI MS studies [9]. FAB, and, later, ESI and MALDI make possible to obtain good quality mass spectra of OS [8,9]. For cyclodextrins, the use of FAB MS [10], and later ESI [11] (named as “ion evaporation atmospheric-pressure ionization mass spectrometry”) has been reported. A year later, an extensive study of electrospray (named as “ion spray”, IS) MS has been done for intact and partially alkylated, partially acylated α-, β-, and γ-CDs using pure solvents and inorganic dopants (salts) in a positive ion mode [12]. In this paper, the first attempt to measure second-order mass spectrum of CD derivative was figured out (as can be seen from Figure 8, bottom, in [12], this attempt almost failed. Recalculation revealed that the α-, not the β-CD derivative was taken, so the figure legend is erroneous). It was found that under electrospray conditions, intact CDs afford adducts with alkaline metal ions (singly charged) or alkali-earth and transition metal ions (multiply charged) along with [M + H]^+^; multiply charged adducts with alkali metal ions were reported for alkylated/acylated derivatives [12]. Indeed, alkali metal ions and protons also give multiply charged ions with intact CDs, for systematic studies including determination of binding constants of CDs with Li^+^, Na^+^, K^+^, Rb^+^, and Cs^+^, see [13]. Ammonium can also form complexes with CDs, thermodynamics of this process was studied using FT ICR MS and molecular modeling [14]. Overlap of peak clusters of the ions bearing different charges (Figure 2) was reported in several papers, e.g. [15,16]. The separation of these ions (for example, [M + H]^+^ and [2M + 2H]^2+^) by mass spectrometry alone is impossible (because of *m*/*z* = n*m*/n*z*), but they are separable by ion mobility spectrometry (IMS) due to their difference in size and shape [17,18].

In MALDI MS, CDs give only singly charged ions. For example, for permethylated β-CD in DHB matrix, only [M + Na]^+^ ion was observed in a special ultrahigh-resolution MS experiment in a positive ion mode [19]. However, there is a pitfall in the interpretation of MALDI MS of some intact and derivatized OSs, especially of CDs, which are prone to form complexes; the phenomenon of association of OSs with DHB matrix was studied [20]. It was found that all intact CDs form adducts [M + DHB − H]^–^ in a negative ion mode and totally or partially methylated β-CDs form adducts [M + DHB + Na (or K)]^+^ in a positive ion mode.

Cyclofructans (CFs), isomers of CDs, are the second class of carbohydrate macrocycles. The most studied of them are hexakis-β-(2→1)-fructofuranose (CF6, **4**), heptakis-β-(2→1)-fructofuranose (CF7, **5**), octakis-β-(2→1)-fructofuranose (CF8, **6**) (Figure 3). In ESI positive ion mode mass spectra, permethylated CFs associate with alkali-metal ions, for CF6, the affinity changes in order of K^+^ > Rb^+^ > Cs^+^ > Na^+^, for CF7, Rb^+^ >K^+^ > Cs^+^ > Na^+^ (solutions in acetone) [21]. Intact CFs also form 1:1 aggregates with alkali metal ions (for extensive ESI MS and DFT molecular modeling studies see [22]). CFs possesses hydrophobic cavities (however, not as prominent as those of CDs) and, hence, are capable of forming inclusion complexes with hydrophobic compounds, e.g., amino acids [23] (compare to CDs: [24,25,26]), so despite a different carbohydrate composition, CDs and CFs have similar properties.

A general problem of all mass spectrometric studies of non-covalent interactions is the adequate evaluation of relations between condensed and gas phases. Otherwise, how to correlate MS peaks intensities registered for ions in the gas phase and concentrations of non-charged molecules existing in the solution which is injected into the ESI ion source? There are many disclaimers in literature cautioning readers to avoid the so-called “mass spectrometric fanaticism”, for example, that it was reported for CD—Trp complexation: all non-1:1 ions observed in ESI MS “are of electrostatic rather than hydrophobic nature” [24]. The discussion of this problem related to comparative study of CDs host-guest complexation was presented in [27] by Zenobi and coworkers. Most papers on MS of CD inclusion complexes are reviewed in [28] and recently in [29] (the latter paper being concentrated on chiral recognition).

The influence of the degree of methylation of CDs on the affinity of alkali metal ions towards CDs was studied in [30]. It was shown for intact β-CD and variously methylated β-CDs that the higher the degree of methylation, the higher is the affinity of Li^+^, Na^+^, and K^+^ towards the CDs; selectivity of bonding increases in order of Na^+^ > Li^+^ > K^+^ in solution and Li^+^> Na^+^ > K^+^ in the gas phase (CID MS^2^ and HCD MS^2^ experiments, DFT molecular modeling).

ESI MS, and especially MALDI MS, now are widely used for profiling derivatized CDs and determination of their degree of substitution (d.s.). In these profiles, the differences in the homologous peaks correspond to specific increments; e.g., for methylated CDs, this is 14 Da (CH_2_). Profiling of charged β- and γ-CD derivatives, both cationic (Me_3_N^+^(CH_2_)^3−^) and anionic (^−^O_3_S(CH_2_)^n−^, n = 2 to 4), was reported for the first time in [31]. Capillary electrophoresis (CE) was taken as an independent method for control, good results were obtained for ESI (IS) and MALDI TOF MS, whereas FAB MS showed inadequate spectra.

### 2.2. Tandem Mass Spectra (MS^n^) of Unsubstituted Cyclooligosaccharides: General Regularities and Attempt of Description

There are several papers devoted to the selective fragmentation of protonated and cationized (metallated) molecules of cyclodextrins (cations) [32,33,34,35,36,37] and deprotonated molecules (anions) [38]. CID MS^2^ spectra of associates of CDs with following cations were studied: Li^+^ [32,34], Na^+^ [33,36], Mg^2+^, Ca^2+^, Cd^2+^, Co^2+^, Cu^2+^, Pb^2+^ [33], Th^4+^, Ce^3+^ [35], NH_4_^+^ [37]. For all cations, singly or multiply charged, the consecutive losses of 162 Da (Glc*p* unit) or, in less extent, for multiply charged ions, losses of 264 Da (C_10_H_16_O_8_) were observed. The former neutral loss occurs definitely due to glycosidic bond(s) cleavage, whereas the latter one apparently involves interresidue cleavage (i.e., the rupture of the C-C bonds in the carbohydrate unit) [33,35]. For cyclofructans CF6 and CF7, formation of [CF – H + Cat]^+^ ions (Cat = Fe^2+^, Co^2+^, Ni^2+^, Cu^2+^, and Zn^2+^) was observed when using aqueous methanol, acetone and acetonitrile solutions for ESI MS; their CID resulted in sequential loss of Fru*f* units (peak difference in 162 Da, the most abundant series), ions of the second (less abundant) series were on 18 Da (H_2_O) heavier [39].

Typical CID MS^2^ spectra (QqTOF instrument, acetonitrile:H_2_O, 50:50 vol. %) of ions generated from α-CD are presented in Figure 4 (here and below, all calculations of *m*/*z* were done by us using Compass Data Analysis 4.0^®^, or an Isotope Pattern application, Bruker Daltonics, Bremen, FRG).

It is impossible to find out “reducing” and “non-reducing” ends for a cyclic oligosaccharide molecule (as well as, in general, there is no beginning/end of any ring), so an attempt to apply the generally accepted Domon-Costello nomenclature of oligosaccharide fragmentation [40] leads to confusion. For example, the authors of [32] referred to [40] and assumed that [M + Li]^+^ and [M + 2Li]^2+^ “fragment by glycosidic cleavages giving B-series of ions...” (note that similar B-series were claimed in [15]), though these fragments (neutral losses of 162 Da, glucose units) may be assigned to Y-type (or Z-type) on the same basis, as reported in the text of the paper [33]: “...we believe that the loss of mass 162 proceeds according to the Y-type ion formation” [33] (p. 1568, left column, also see Scheme 1, p. 1570 reprinted here as Figure 5). The same Domon and Costello paper [40] was referenced. In contrast to the above suggestions, the authors in the same paper [33] introduced new designations of A, B, C, and D-series for fragmentations of doubly charged cations [M + Cat]^2+^ (Cat is divalent metal cation, QqTOF mass spectrometer, collision energy, c.e. 25–35 eV) in the table of the MS^2^
*m*/*z* data and presented a description of them below the table. Briefly, A-series is formed by doubly charged ions differed in 81 Th. B-series arises due to charge separation reaction due to loss of [Glc_n_ + H]^+^ (difference in 162 Da). Low abundant C-series is formed by singly charged ions smaller than B-peaks on 264 Da; this cleavage is accompanied by C-H hydrogen transfer (demonstrated by H/D isotopic exchange in OH groups). Low abundant D-series may be regarded as a result of loss of H_2_O from B-series ions (not observed for Co, Cu, and Pb additives). For a representative MS^2^ spectrum, see Figure 6, one can conclude that this is an ad hoc classification poorly related to the Domon-Costello nomenclature; only “B-ions” has a weak similarity to B-series stated in [40]. Unfortunately, the survey of the literature has shown that there is no adequate, general nomenclature to describe the cleavages for cyclic oligosaccharide ions (see Section 2.5).

### 2.3. The Difference in Mass Spectra of Cyclic and Acyclic Oligosaccharides

#### 2.3.1. FAB MS, MALDI TOF MS, and ESI MS of Cyclic Oligosaccharides

One of the applications of MS to structure elucidation of OSs is the determination of their acyclic (the most common) or cyclic (more rare) nature. Of course, acyclic or cyclic OS may be branched or carry side chains. Cyclic OS are lighter than the corresponding acyclic OS by a definite increment; for underivatized OSs, this is H_2_O (*ca*. 18 Da), for permethylated OSs, C_2_H_6_O (*ca*. 46 Da), for peracetylated OSs, C_4_H_6_O_3_ (*ca*. 102 Da), etc. Note, that the MS of acyclic OS having an anhydro unit in its composition looks like that of cyclic one due to their isomerism. Some structural studies based on MS (along with NMR) are presented below. A previously unknown glucan (classified by its function as osmoregulated periplasmic glucan, OPG) was isolated from a recombinant strain of a *Rhizobium meliloti* TY7, an *ndv*B mutant carrying a locus specifying β-(1→3, 1→6) glucan synthesis from *Bradyrhizobium japonicum* USDA110, using reversed phase chromatography [41]. The compound was of carbohydrate nature free of phospholipid moieties usual for Rhizobia. FAB MS of the oligosaccharide showed peaks at *m*/*z* 1945 and *m*/*z* 1967, which were assigned to [M + H]^+^ (calcd. *m*/*z* 1945.6) and [M + Na]^+^ (calcd. *m*/*z* 1967.6), respectively (M = Glc_12_). Acyclic dodecahexaose should be heavier by 18 Da, so, this one had likely a cyclic structure. ^1^H- and ^13^C-NMR spectra were rather complex, but they demonstrated the presence of mainly β-(1→3)-linked (definitely not of α-configuration) glucose residues along with a minor β-(1→6)-glycosidic bond. The first Smith degradation afforded the product consisting of eleven hexose units (FAB MS *m*/*z* 1783, [M + H]^+^, calcd. *m*/*z* 1783.6, and *m*/*z* 1805, [M + Na]^+^, calcd. *m*/*z* 1805.6). The second Smith degradation gave the product consisting of only β-(1→3)-Glc*p* units according to ^1^H- and ^13^C-NMR spectra (i.e., this is cyclodekakis-(1→3)-β-glucosyl), which was resistant to the next Smith degradation. Surprisingly, the MS of this carbohydrate expected to be cyclodecalaminarinose exhibited the main peak at *m*/*z* 1735, which the authors assigned it to [M + 5Na]^+^, which is definitely unbelievable. A more reasonable explanation of this peak is complexation of the analyte with glycerol usually used as a FAB matrix (although thioglycerol and *m*-nitrobenzoic acid were reported as matrices in the Experimental part), for [Hex_10_ + Gro + Na]^+^, calculation gives *m*/*z* 1735.56, so the structure of the starting material was assigned to β-(1→6)-laminaro(cyclodekakis-β-(1→3)-glucosyl). Later, the same authors isolated the both cyclic OSs (branched, cyclic dodecasaccharide and cyclic decasaccharide) from different mutants of *Bradyrhizobium japonicum* [42]. At the same time, an extensive FAB and MALDI MS study of laminarans (reserve β-(1→3, 1→6)-glucans from brown algae) was performed [43]. It was found for laminarans from *Chorda filum* and *Cystoseira crinita*, that some peaks in MALDI MS of these native laminarans were accompanied by “minus 18 Da” satellites and the corresponding peaks in MALDI MS and FAB MS were accompanied by “minus 46 Da” ones. Because other laminarans (both G-chains, i.e., with free reducing end, and M-chains, terminated with β-(1→1)-linked mannitol) did not demonstrate a loss of water under MALDI conditions, the authors proposed the occurrence of macrocyclic structures in these laminarans. Since no experiments to find out independent proofs were done, the precaution “Whether the cyclic components are macrocycles or have an anhydro sugar at the reducing end remains to be established” was stated [43]. This remains true until now.

Similar studies were carried out for OPGs from several agriculturally important microorganisms (symbiotic, *e.g*., Rhizobia, nitrogen-fixing bacteria, or parasitic, like *Xantomanas campestris*, responsible for black rot of cruciferous plants). An earlier application of FAB MS done by Dell, was to establish cyclic structures of β-(1→2)-linked glucans (d.p. from 17 to 24, cyclosophoraoses) isolated from Rhizobia and Agrobacteria. It was shown by ^13^C-NMR that these cyclic structures are homogeneous (β-(1→2)-Glc*p* units only) and unbranched [44]. Later, two other cyclic OPGs were isolated from *Bukholderia solanacearum* and *Xantomonas campestris pv. citri* [45]. Their cyclic structures were demonstrated successfully by MALDI TOF MS: the main peaks both assigned to [M + Na]^+^, were at *m*/*z* 2129.7 (calcd. for [Glc_13_ + Na]^+^
*m*/*z* 2129.67, *B. solanacearum*) and *m*/*z* 2616.8 (calcd. for [Glc_16_ + Na]^+^
*m*/*z* 2616.84, the second peak in isotopic cluster having 100% abundance, *X. campestris*), respectively. The ions [M + K]^+^ were mentioned with no data. Heterogeneity of these cycles (the presence of one β-(1→6)-Glc*p* units inside each cycle) was approved by NMR spectrometry (^1^H- and ^13^C-NMR, including 2D procedures) and molecular dynamics calculations. Cyclic OPGs usually bear side chains, e.g., phosphatidylglycerols giving a +C_3_H_7_O_5_P increment (*ca*. 154 Da each) [46]. In this paper, two glycerophosphorylated α-cyclosophorohexadecaoses (M_1_, monophosphorylated, M_2_ diphosphorylated) were isolated from a different stain of *X. campestris* and characterized by MALDI MS in a positive ion mode and by ESI MS in a negative ion mode. For MALDI MS (DHB matrix), the following data were presented: *m*/*z* 2792.7, [M_1_ – H + 2Na]^+^, calcd. *m*/*z* 2792.8 (for all, the most abundant second peak in the cluster), *m*/*z* 2969.1, [M_2_ – 2H + 3Na]^+^, calcd. *m*/*z* 2968.80. In ESI MS, the main triply and doubly charged ions were observed: *m*/*z* 915.9, [M_1_ – 3H]^3−^ (calcd. *m*/*z* 914.9433); *m*/*z* 967.7 [M_2_ − 3H]^3−^, (calcd. *m*/*z* 966.2777); *m*/*z* 1374.6, [M_1_ − 2H]^2−^ (calcd. 1372.42), *m*/*z* 1451.6, [M_2_ − 2H]^2−^ (calcd. 1449.92). There is no explanation why the exp./calcd. data coincidence for ESI MS was not so good as for MALDI TOF MS, possibly, this due to low resolution of the triple quadrupole analyzer used in this work. No MS^2^ data were reported.

The appropriate side chain (e.g., possessing specific affinity) can be introduced artificially. Biotinylation through amidocaproyl spacer was done by Cho et al. [47]. A mixture of cyclooligosophoraoses from Rhizobia (d.p. 17 to 22) were selectively monotosylated at C-6, subjected to azidation followed by reduction with PPh_3_ in DMF, and the purified amine was acylated with biotinamidohexanoic acid *N*-hydroxysuccinimide ester. MALDI TOF MS (DHB matrix) of monoaminated and monobiotinylated cyclooligosophoraoses displayed prominent [M + Na]^+^ peaks for d.p. 17 to 22 which showed good coincidence with the calculated values. As expected, the difference between the corresponding peaks with the same d.p. is equal to ca. 339 Da (the total mass of biotin moiety and caproyl linker, C_16_H_25_N_3_O_3_S, 340 Da was reported in the paper).

#### 2.3.2. Cyclic and Acyclic Oligosaccharides: Differences in Affinities to Metal Cations

One can suppose than there are differences in the affinities to alkali metal cations (or other cations) between linear and cyclic oligosaccharides. In fact it was shown by Choi et al. [48] that the relative ion intensities of α-CD and β-CD associates with Na^+^, K^+^, and Cs^+^ (not H^+^) in MALDI MS were much higher that of corresponding ionized maltohexaose and maltoheptaose, respectively; differences showed an increasing trend with the radii of metal cations, and β-CD had higher ionization efficiency than α-CD. However, no proofs of general character of this regularity were demonstrated and even if such proofs will be found, it seems problematic to use this phenomenon in assignment of isomeric acyclic and cyclic OSs without authentic or reference samples (*cf*. [43]).

#### 2.3.3. Cyclic and Linear Oligosaccharides: Differences in Tandem Mass Spectra

To use fragmentation techniques to distinguish linear and cyclic OSs is promising in order to avoid isolation of individual compounds. The first attempt to realize this approach was done by P.J. Derrick and colleagues [49]. Dextran, maltodextrin, and γ-CD were subjected to in-source decay (ICD) under MALDI MS. Of course, these gluco-OSs have different linkages and different secondary structures, nevertheless, the distinction in degrees of fragmentation clearly correlated with degree of freedom in saccharide chains (determined for solutions by NMR). For γ-CD, fragmentation was negligible even at elevated laser power in contrast to acyclic OSs, especially dextran which was fragmented the most. Maltodextrin had an intermediate degree of fragmentation.

Later, a similar regularity was demonstrated for ESI MS in more sophisticated experiment using CID, PQD, and HCD activation procedures for three pairs of α-CD/maltohexaose, β-CD/maltoheptaose and γ-CD/maltooctaose [50]. A LTQ orbitrap hybrid instrument was used and [M + Na]^+^ and [M + H + K]^2+^ ions were fragmented for each compound. It was shown that three methods of activation are complementary. For CDs, all of the fragments were formed by glycoside bonds rupture, no cleavages of residues were observed. Comparison of CE_50_ demonstrated that slightly higher collision energy is needed to cleave CD than the related linear maltooligosaccharide (of course, the corresponding ions should be considered). Though the authors claim that CID/HCD combination opens a way for deciphering of structures of complex carbohydrates “alone or in mixture”, we do not share their optimism because the observed effects seem very tiny.

Thus, the choice between acyclic or cyclic structure can be supported by ESI or MALDI MS if a small amount (enough for an NMR study) of rather pure compound is isolated, but such a choice for a component in a complex mixture (when only MS alone can be used) remains problematic despite the great progress of tandem techniques during the last thirty years.

### 2.4. Complexation of Cyclic OS with Other Molecules And Guests’ Influence on CDs Induced Decay

As mentioned above, most of the papers on mass spectrometry of cyclodextrins are concerned with host-guest complexes and determination of their stoichiometry, thermodynamics and chiral recognition; these aspects were thoroughly reviewed previously [28,29] and are beyond the scope of this review. Nevertheless, the problem of mutual host/guest influence should be discussed. The general principle of these effects was formulated by Lebrilla and coauthors [51]. Under CID conditions, inclusion complexes can: (A) dissociate with charge retention on the host (A1) either in the guest (A2) molecule; or (B) result in cleavage of the host oligosaccharide (CD), or (C) result in cleavage of the guest molecule. CID MS of the complexes of permethylated β-CD and native CD with adenine, cytosine, guanine, thymine, and four corresponding deoxy nucleobases were studied using FT-ICR or ion trap mass spectrometers. For comparison, maltoheptaose and permethylated maltoheptaose were taken. It was shown that the preferred dissociation pathway depends on the structures of both the host OS and guest molecule. For example, cytosine assisted in fragmentation via path B inducing ring opening and intramolecular proton transfer; the presence of amino group (or, more definitely, both basic and acidic groups) in the guest molecule seemed substantial. A similar mechanism was proposed independently for fragmentation of a complex of the protonated β-CD—5-methoxytriptamine (5MTA) also studied by CID MS^2^ [52]. The authors have observed formation of β-CD fragments [(Glc)_n_ + H]^+^, n = 2–5 by activation of the protonated complex (path B according to [51], see above) and proposed the relation of this fragmentation with deamination of the guest (FAB or IS, triple quadrupole, or ESI, ion trap) [52], because of proton transfer from ammonium to β-CD was not observed previously; ammonium is assumed to be more stronger acid than protonated 5MTA. The authors proposed that deamination of 5MTA is the primary process and then the excited deamination product molecule (activated guest) transfer proton to the host (β-CD); after that the excited host β-CD undergoes decay (elimination of protonated guest is a parallel process). In both studies, intramolecular guest—host proton transfer was not supported in direct labeling experiment which seems really difficult to create avoiding possible side effects and uncertainties in interpretation.

It necessary to mention that not only protonated complexes of CDs were studied by tandem MS. For example, it was shown that [β-CD + PhMe + Fe]^2+^ complex is substantially more stable than [β-CD + PhMe + H]^+^; the same effect was observed by adding Mg^2+^ [53]. Other aromatic, low-polar molecules were studied as guests, and the stabilization effect was also observed whether real inclusion complex or electrostatic associate were formed. In this case, CID fragmentation proceeded via path A1, no CD fragments were observed. Thus, the guest-host effects are versatile (from negligible to strong) and the influence of associated cation is necessary to take into account.

### 2.5. Determination of Positions of Substituents in Cyclic Oligosaccharides—General Considerations and Problems

At present time, there are various derivatives of cyclic OSs, especially CDs prepared for different purposes. First-order MS (EI, MALDI) can obviously applied for profiling and determination of d.s. [31,37]. Since the regioisomers have the same *m*/*z* of their ions, the application of MS^2^ is required. Though the differences of MS^2^ of regioisomeric cyclic OSs seem to be predictable, several studies were done; their results revealed some problems (see below).

#### 2.5.1. Tandem Mass Spectra of Monosubstituted Cyclic Oligosaccharides: The Effect of a Substituent

At first, the most used selective modification of CDs is *O*-6-tosylation of Glc*p* unit [54]. Then the OTs group may be converted to an amino group by azidation followed by reduction or substituted by a nucleophile, for example, an alkyl diamine [54]. By manipulation of protecting groups, other hydroxy groups in glucopyranosyl residues in CDs may be interchanged for amino groups. The amino groups may be subsequently modified by selective *N*-acylation or *N*-alkylation thus opening a route to various derivatives for versatile applications. In this section, we consider amino-CDs and acylamido-CDs as “substituted CDs”, though they may be regarded as “mixed cyclic Oss” (containing amino sugars, see Section 2.7). The effects of amino and acetamido functional groups on ionization and fragmentation of carbohydrate units in modified β-CDs were studied in [55]. Monosubstituted 3-NH_2_-β-CD **7** and 3-NHAc-β-CD **8** (Figure 7) were taken for the tandem MS (LID MALDI and ESI MS^2^) study. Note that a synthesis of the derivatives resulted in inversion of the C-2 and C-3 configurations, so the glucopyranose residue transformed into 3-amino-3-deoxyaltropyranose unit (designated here as Hex*N*), however, one can believe that the change of stereochemistry had an apparently negligible effect on tandem mass spectra. For 3-NH_2_-β-CD, both [M + H]^+^ and [M + Na]^+^ were observed in MALDI and ESI MS. LID and CID of the protonated molecule resulted in similar secondary-order mass spectra with predominant formation of aminohexose-containing fragments, so, a proton is definitely localized on the amino group and the first cleavage of glycosidic bond occurs near the Hex*N* residue (Figure 8b). For LID MALDI TOF/TOF MS of [M + Na]^+^, Hex*N*-containing fragments were only [Hex*N* + Glc + Na]^+^ and [Hex*N* + Glc_2_ + Na]^+^ along with [Glc_n_ + Na]^+^, n = 2 and 3, no aminosugar-containing fragments were observed for n = 4, 5, and 6 (Figure 8a). One can assume that Na^+^ ion coordinates both with Glc and Hex*N* residues thus results in almost random cleavage of glycosidic bonds. In contrast, in LID MALDI MS of [M + Na]^+^ for 3-NHAc-β-CD, Hex*N*Ac-containing fragments are at least twice more abundant than corresponding [Glc_n_ + Na]^+^ ions (Figure 9). The authors suggested that Na^+^ is prone to coordinate with Hex*N*Ac residue stronger than with underivatized Glc unit [55].

#### 2.5.2. Tandem Mass Spectra of Disubstituted CDs: Effect of Positions of Substituents

The influence of the positions of regioisomeric, disubstituted β-cyclodextrins on their ESI MS^2^ mass spectra was studied in [15] for two triplets of substituted β-cyclodextrins: bis(6-bis(2-picolyl)glycylamido)-6-deoxy-β-CD (**9a**–**11a**) and bis(6-*O*-tosyl)-β-CD (**12**–**14**). One can assume that mass spectra of decay for regioisomers of modified CDs (as well as for any cyclic oligosaccharides) may differ from each other since a double rupture of glycosidic bonds can result in different combinations of primarily bonded units. A critical consideration of this approach may be formulated in two questions:(1)Are the bonds between substituents and the carbohydrate units strong enough in comparison to glycosidic bonds, and(2)Do transitions of substituents between carbohydrate residues (i.e., rearrangements) take place?

The negative answer on the first question and/or positive on the second one makes mass spectrometric structural assignment of these compounds impossible. A study of two series [15] (see above) has shown that both variants are possible. Thus, for regioisomers with bispicolylamide substituents (AB has adjacent substituted units, **9a**, in AC, they are separated with one and four unsubstituted units, **10a**, and in AD, **11a**, substituted units are separated with two and three unsubstituted units, Figure 10) fragmentation of glycosidic bonds is predominant and transition of these substituents between units does not observed thus resulting in characteristic MS^2^ for all three [M + H]^+^ ions (these peak series are basically arisen due to a cleavage of glycosidic bonds, Figure 11). For AB, the fragment consisting of two substituted units is present (*m*/*z* 801), but it is *absent* in MC^2^ of AD and AC. The fragment consisting of two substituted and one unsubstituted unit (*m*/*z* 963) is *absent* в MS^2^ of AD. The peak of the ion resulting from elimination of one substituent (*m*/*z* 1732.5) is present in MS^2^ of all three regioisomers, nevertheless, this pathway does not dominate. For bis-6-O-tosylates, the fragmentation differs drastically: CID MS^2^ of [M + H]^+^ for ditosylates AC and AD are practically the same, and for AB, the same pattern of ions was observed though peaks assigned to ditosylated maltose (*m*/*z* 633) and ditosylated maltotriose (*m*/*z* 795) have higher relative abundances than those of AC and AD (*m*/*z* 479, 100 % is assigned to monotosylated maltose, Figure 12). Apparently, *O*-tosyl group can migrate between glucosyl residues in cyclodextrin under CID. Unfortunately, it was not shown in [15], what collision energy (“35 or 70 eV” was reported in Experimental with no further reference) was definitely applied in each case: that is likely to be critical for the result of tandem experiment. Moreover, there are no tandem spectra for [M + Na]^+^ tosylates: such an investigation may be promising because it is generally accepted that the alkali metal-associated ions of OSs do not undergo rearrangements [56,57].

It is difficult to discuss a similar early study [36] because the data obtained there look unreliable, at least, in part. Because a monoquadrupole, low resolution instrument was available, unselective CID was achieved by applying an elevated cone voltage (190 V). Nevertheless, CID pseudo-MS^2^ of [M + Na]^+^ of three 6,6′-diamino β-CDs **9b**–**11b** seem to be conclusive. Really, in the range of sodiated three-residues ion (*m*/*z* 507 to *m*/*z* 511, Figure 13) the differences between AB, AC, and AD isomers were clear despite of incomplete resolution of the peaks: the absence *m*/*z* 507 peak for AD, *m*/*z* 509 was the highest for AB, and for AC, comparable intensities of *m*/*z* 508 and *m*/*z* 509 were observed (the hypothesis of random cleavage of glycosidic bonds in [M + Na]^+^ was later proved for a monoamino derivative [55], see above). Three isomeric tosylates and two isomeric mesitylates were studied without independent determination of their structures hence the assignments in the second part of the study cannot be considered as reliable.

### 2.6. Tandem Mass Spectra of Complex And Highly Substituted CD Derivatives

#### 2.6.1. Tandem Mass Spectra of CD Derivatives Possessing Succinyl-Spacered Dicationic Substituent (Gemini Surfactants)

Cyclic OSs, especially CDs, are used as starting compounds for the construction of complex organic and hybrid molecular systems. For example, CDs are applied for preparation of drug delivery agents in which a CD core is tethered by ether or ester bonds with linkers bearing side ionogenic groups. A scheme of CID fragmentation of a series of surfactants based on 6-*O*- or 6-*N*-monoacylated β-CD (Figure 14) was presented [58]. The study was performed using an ESI-equipped hybrid triple quadrupole—linear ion trap (LIT) instrument. For all of four β-CD dication derivatives, abundant M^2+^ ions were observed. CID of M^2+^ resulted in the loss of one *N*-alkyl (in a form of alkene) followed by NHMe_2_ loss. The [M − (NMe_2_R)]^2+^ ion eliminated one singly charged Glc residue from β-CD thus opened the cycle. Two pathways of further fragmentation were realized. Path 1: Glc neutrals are sequentially lost. Path 2: the second alkyl and NHMe_2_ were lost followed by sequential loss of Glc units. Cleavage of the carbohydrate residues also occurred simultaneously. This complicated scheme of thoroughly investigated fragmentation was supported using MS^3^ to reveal fragment ion transitions (and for one compound, deuterium labeling of alkyls were applied). All studied compounds had similar fragmentations, no drastic differences between *O*- and *N*-succinates were reported.

#### 2.6.2. Tandem Mass Spectra of Polysubstituted (Multiply Acylated) CD Derivatives

For the last two decades, CDs were used as scaffolds for star polymers, the best known being the oligolactide-tethered CDs produced by ring-opening polymerization of lactide (the cyclic dimer of lactic acid) initiated by CD, where oligolactide forms a chain attached to the CD [59,60,61,62,63,64]. To optimize this synthesis and to achieve high d.s., it is necessary to control the polymerization; both NMR and MS are applied (sometimes along with other methods, e.g., IR spectroscopy and electron microscopy [64]). The most often used MS procedure is MALDI MS profiling [59,60,62,63]. In the recent work of Peptu and Mosnaček collaboration [63], a simple formula was reported for MALDI profiling of a fraction of β-CD-LA:*m*/*z* = 1134 (β-CD) + n × 144(LA) + 39 (K)(1)
where n is d.s., LA means an increment of the residue of lactide (144 Da), K means that potassium adducts are mainly formed, however, adducts with Na^+^ were also identified. It was stressed that “LC with on-line ESI MS detection has problems with the analysis of polymer species related to formation of multicharged ions thus creating difficulties in spectra interpretation, especially in the case of polydisperse oligomer mixtures” [63]. To solve this problem, Peptu et al. studied the possibility of using tandem mass spectra for the structure elucidation of CD esters [65,66]. Totally acetylated β-CD (TABCD), β-CD randomly acylated with butyrolactone (HBCD) [65], and β-CD partially substituted with oligolactide chains [66] were used (Figure 15). Tandem mass spectra were acquired with a QqTOF instrument, elevated cone voltage (200 V) and activation in the collision cell were applied simultaneously. The accuracy of the *m*/*z* measurements may be considered as fair, the integer monoisotopic masses have good agreement with those calculated. In the MS^1^ of TABCD, [M + H]^+^ and [M + Na]^+^ were present, and their MS^2^ spectra differed drastically. For MS^2^ of [M + H]^+^ (c.e. 20 eV), stepwise losses of HOAc and Glc units were observed: after elimination of four HOAc molecules, elimination of one Glc unit occurred until all co-monomer units were depleted. This process was readily illustrated in the table of all possible *m*/*z* values, where really observed *m*/*z* ones were highlighted forming a kind of ladder. For MS^2^ of [M + Na]^+^ (c.e. 120 eV), only consecutive losses of four molecules of HOAc (accompanied in lesser extent, by C_2_H_2_O losses) were observed. In a small *m*/*z* range, abundant peaks corresponding to acetylated Glc unit fragments were present. As it was demonstrated previously by NMR and MS [67], the applied sample of HBCD contained β-CD esterified by single hydroxybutyryl residues (mainly at O-3), their numbers varied from one to seven. The aim was to prove by MS^2^ alone that there were no oligomeric hydroxybutyryl substituents (which may occur due to esterification of a hydroxyl to give a hydroxybutyryl moiety). Abundant [M + Na]^+^ ions along with quite visible [M + H]^+^ ones were observed for each component in ESI MS. For MS^2^ of [M + H]^+^ of 5HB-substituted β-CD (c.e. 15 eV), a subsequent loss of Glc units (162 Da) accompanied by losses of HB substituents (104 Da) and their anhydro forms (86 Da) occurred. A similar "ladder", but with smaller steps was pictured in the table created analogously to MS^2^ of TABCD. MS^2^ of the [M + Na]^+^ ion (c.e. 110 eV) contained peaks corresponding to different fragmentation pathways. Ions *m*/*z* 1483 and *m*/*z* 1379 are formed due to losses of one and two HB molecules. The ion *m*/*z* 1425 resulted in loss of one Glc unit, and the ion *m*/*z* 1339 was formed due to (Glc-HB) loss (104 + 86 Da). Because of the ion *m*/*z* 1397 (loss of HB_2_, 104 + 86 Da) was not observed, one can conclude that there were likely no di-HB substituent attached to β-CD. The origin of the quite abundant *m*/*z* 1543 ion (loss of 44 Da) was not definitely explained by the authors; due to low resolution, it was impossible to find out what neutral (C_2_H_4_O or CO_2_) was eliminated. The above fragmentation pathways classified by Peptu et al. are presented in Figure 15. Note, that this is an ad hoc description, and the letters A, B, and C have different meaning from that of Domon-Costello nomenclature [40].

Recently, a mixture of derivatized β-CDs partially substituted with oligolactide chains was studied by GPC, NMR, and MS [66]. From the NMR spectra the authors concluded that oligolactides were attached as multiple short chains to different OH groups of the CD molecules. From MALDI MS profiling, an average of 15.6 lactate units per β-CD were calculated. Tandem MS (ESI MS^2^ and MALDI LID) were thoroughly described for the [β-CD-LA_4_ + Na]^+^ ion with total assignment of composition of fragment ions. In contrast to the above examples, multiply repeated units in substituents capable of fragmentation were definitely present, so three other pathways were observed designated as G (loss of substituted Glc unit) and E1 and E2 (fragmentation of side lactide chains), Figure 15, bottom. As expected, the most intense peaks corresponded to fragments containing an even number of lactate units. Minor peaks with an odd number of lactate units were also present because transesterification is possible during ring-opening, CD-induced polymerization of lactide. The same regularities were found for K^+^ and Li^+^ adducts; heavier ions having more lactate units gave homologous fragmentation patterns.

Several related studies on multiply acylated CDs were reported. Profiling of HBCD mixture (see above) was done successfully using HPLC-ESI-MS along with NMR of separated fractions [67]. A similar study of modified, amphiphilic CD mixtures was done with HPLC-MS [68]. Selectively *O*-benzylated (with only two free OH groups) β-CD was transferred to a homologous mixture of oligopolylactides, which was characterized by MALDI TOF profiling along with NMR spectrometry [69].

One can mention that the aim of these mass spectrometric studies [65,66] was deliberately restricted: only stoichiometry of CDs derivatives was correlated with MS^2^ profiles, not positions of substituents, so the problem of acyl migration was beyond the scope. Despite the complexity of the NMR spectra of derivatives of CDs, this method including 2D NMR procedures remains the gold standard in structure determination of substituted CDs.

### 2.7. Tandem Mass Spectra of Cycloaminooligosaccharides and Mixed Cyclooligosaccharides

#### 2.7.1. General Information

Recently, Chizhov et al. reported extensive studies concerning high resolution tandem mass spectrometry (ESI QqTOF) of cyclooligo-β-(1→6)-d-glucosamines, cyclooligo-β-(1→6)-d-*N*-acetyl-glucosamines (from two to seven Glc*N* or Glc*N*Ac residues, respectively, **19a**,**b**–**24a**,**b**, Figure 16) [70], and mixed isomeric β-(1→6)-linked cyclic tetrasaccharides of the Glc*p*_2_Glc*pN*_2_ composition (**25** and **26**, Figure 17) [71]. Synthetic details for the preparation, structural and conformational analysis of cyclo-β-(1→6)-d-glucosamines **19a**,**b**–**24a**,**b** were described in [72,73,74].

These compounds were used as versatile scaffolds in the design of oligodentate blockers of oligomeric bacterial lectins [75,76] and building blocks for creation of artificial ion channels [77,78].

First-order, electrospray ionization (ESI) mass spectra of compounds **19a**,**b**–**24a**,**b**, **25** and **26** have prominent peaks of ions [M + H]^+^, [M + 2H]^2+^, [M + Na]^+^, [M + K]^+^, and a moderate peak of an [M − H]^−^ anion. The ions [M + NH_4_]^+^ are formed only for cyclic *N*-acetylglucosamines **19b**–**24b**, not for cyclooligoglucosamines **19a**–**24a**.

#### 2.7.2. Tandem Mass Spectra of Cycloaminooligosaccharides

In the positive ion mode ESI CID MS^2^ spectra of cyclooligo-β-(1→6)-d-glucosamines **19a**–**24a** and cyclooligo-β-(1→6)-d-*N*-acetylglucosamines **19b**–**24b**, cleavages of the glycosidic bonds were the principal processes, they are accompanied by eliminations of molecules of water (and ammonia for cyclooligo-β-(1→6)-d-glucosamines) [70]. For example, protonation of the cyclic dimer of glucosamine **19a** produced the [M + H]^+^ and [M + 2H]^2+^ ions. The main fragment of activation (c.e. 25 eV) of the singly charged ion, [Glc*N* + H]^+^ (Figure 18) had the same *m*/*z* 162.0760 that the doubly charged ion, but it differed from that one by positions of isotopic peaks. This fragment arose by cleavage of two glycosidic bonds. Peaks with lower intensities resulted in further fragmentation of the glucosamine residue with a loss of one or two molecules of water, [Glc*N* + H − H_2_O]^+^ and [Glc*N* + H − 2H_2_O]^+^, respectively. The protonated molecule can eliminate small fragments, one molecule of ammonia [Glc*N*_2_ + H − NH_3_]^+^, one molecule of water [Glc*N*_2_ + H − H_2_O]^+^, or both molecules, [Glc*N*_2_ + H − H_2_O − NH_3_]^+^. The formation pathways of the middle-range ions of low abundance (lower than 10 %) was described as a result of rearrangement. Peak at *m*/*z* 203.1023 (marked in Figure 18 by an asterisk) had a composition of C_8_H_15_N_2_O_4_, i.e., [Glc*N*_2_ + H − C_4_H_8_O_4_]^+^; its formation was described as a consequent cleavage of one glycosidic bond followed by elimination of the C3-C6 fragment of the second residue (charge-remote elimination). The peaks at *m*/*z* 185.0919, [Glc*N*_2_ + H − C_4_H_8_O_4_ − H_2_O]^+^ and *m*/*z* 167.0814 [Glc*N*_2_ + H − C_4_H_8_O_4_ − 2H_2_O]^+^ was explained analogously (further elimination of one or two molecules of water, respectively). The peak at *m*/*z* 180.0861 was assigned to the [Glc*N* + H + H_2_O]^+^ ion. The origin of peaks at *m*/*z* 174.0756 and *m*/*z* 186.0761 (one and two carbon atoms more than in [Glc*N* + H]^+^, respectively) was difficult to understand, possibly they occurred due to unknown skeletal rearrangements different from those known before (loss of internal residues and/or migration of a terminal residue) [57]. In this case, one can suppose that the ions at *m*/*z* 186 and *m*/*z* 168 were formed due to the transfer of the CH_2_CO fragment from one unit to the amino group of the other one followed by cleavage of the glycosidic bond and the loss of one or two molecules of water (in comparison, fragmentation of the protonated molecule of **19b** resulted in quite intense peaks of these ions). Activation of the [M + 2H]^2+^ ion (c.e 10 eV) resulted in appearance of the main peak at the same *m*/*z*, but differed from the residual signal by the position of isotopic peaks (see above) along with smaller peaks of the [Glc*N* + H − H_2_O]^+^ and [Glc*N* + H − 2H_2_O]^+^ ions (Figure 19). The peak at *m*/*z* 102.0550 possessed the composition of C_4_H_8_NO_2_^+^, i.e., it arose from elimination of the neutral fragment of C_2_H_4_O_2_ from the glucosamine residue. Small peaks in the range of *m*/*z* from 162 to 203 are the same as in MS^2^ CID for the [M + H]^+^ ion, but additional peaks were observed: intense peak at *m*/*z* 190.0707 (C_7_H_12_NO_5_^+^, i.e., in CO larger than protonated glucosamine residue) and the small peak at *m*/*z* 204.0871 (C_8_H_14_NO_5_^+^ that corresponds to [Glc*N*Ac + H]^+^, see below). The CID MS^2^ of the [M + Na]^+^ ion was not described due to its low intensity.

Mass spectra of the negatively charged ions of the studied compounds demonstrated low abundant deprotonated molecules [M − H]^−^. As an example, the CID MS^2^ (c.e. 20 eV) of the ion [M − H] ^−^ of **20a** was considered. The decay of this anion differed from that of protonated molecule because the main fission occurred not at glycosidic bonds. The small peak at *m*/*z* 160.0612 can be considered as a deprotonated link of glucosamine [Glc*N* − H]^−^, i.e., the product of cleavage of two glycosidic bonds, nevertheless, the ion [Glc*N*_2_ − H]^−^ was not observed. The ions of lower masses occurred due to elimination of molecules of ammonia (*m*/*z* 143.0361, C_6_H_7_O_4_^−^) and water (*m*/*z* 142.0508, C_6_H_8_NO_3_^−^), and the ion at *m*/*z* 101.0247 (C_4_H_5_O_3_^−^) was possibly formed by elimination of the neutral C_2_H_5_NO moiety that may correspond to the O(1)-C(1)-C(2)-N(2) fragment. The main fragment peak at *m*/*z* 220.0831 (C_8_H_14_NO_6_^−^) can be formed only as a result of a cleavage of the C—C bond (presumably C(4)-C(5) and C(5)-O(5) of one of the residues with the retention of one (1→6)-glycosidic bond and the rupture of another one). Small peaks at *m*/*z* 464.1906 and *m*/*z* 446.1797 corresponded to elimination of one or two molecules of water. The peak at *m*/*z* 423.1628 (C_16_H_27_N_2_O_11_^–^) may be a result of elimination of the O(1)-C(1)-C(2)-N(2) fragment of one unit. The further elimination of the C_2_H_4_O_2_ fragment resulted in formation of *m*/*z* 363.1435 (C_14_H_23_N_2_O_9_^−^) anion. The ion at *m*/*z* 280.1053 (C_10_H_18_NO_8_^−^) corresponded to the formal addition of the C_4_H_8_O_4_ fragment to the deprotonated glucosamine residue. Formation of this ion was proposed through the rupture of C-C and C-O bonds in two glucosamine residues with the retention of the third one as a core. It was concluded that primary fragmentation of the negatively charged ions (deprotonated molecules) proceeds in a different way than that of positively charged ones [70].

Mass spectra of cyclic oligo*-N*-acetylglucosamines were similar to those of their acyclic analogs. For example, MS of cyclic dimer of *N*-acetylglucosamine **19b** possessed peaks of [M + H]^+^, [M + NH_4_]^+^, [M + Na]^+^, and [M + K]^+^ ions. CID MS^2^ of [M + H]^+^ ion resulted in a cleavage of glycosidic bonds and formation of the ion at *m*/*z* 204.0865 ([Glc*N*Ac + H]^+^); under c.e. of 10, 15, and 20 eV, the peak of this ion was the most intense. Elimination of one, two, and three molecules of water resulted in formation of ions at *m*/*z* 186.0761, *m*/*z* 168.0655, and *m*/*z* 150.0550, respectively. At c.e. 40 eV, the peak at *m*/*z* 138.0549 (C_7_H_8_NO_2_, [Glc*N*Ac + H − H_2_O − CH_2_O]^+^) became the main one. Two other intense peaks (c.e. 40 eV) at *m*/*z* 144.0654 (C_6_H_10_NO_3_) and *m*/*z* 126.0549 (C_6_H_8_NO_2_) were assigned to the fragment ions [Glc*N*Ac + H − C_2_H_4_O_2_]^+^ and [Glc*N*Ac + H − C_2_H_4_O_2_ − H_2_O]^+^, respectively. Fragmentation of the ion [M + NH_4_]^+^ (c.e. 20 eV) was similar to that of [M + H]^+^. In the range of higher *m*/*z*, the products of elimination of one ant two molecules of water, *m*/*z* 389.1544 and *m*/*z* 371.1451 were found. In CID MS^2^ (c.e 50 eV) of metallated molecules of **19b**, only cleavages of two glycosidic bonds were revealed ([Glc*N*Ac + Na]^+^, *m*/*z* 226.0686 and [Glc*N*Ac + K]^+^, *m*/*z* 242.0355, respectively), and elimination of H_2_O from these fragments resulted in [Glc*N*Ac + Na − H_2_O]^+^, *m*/*z* 208.0575 and [Glc*N*Ac + K − H_2_O]^+^, *m*/*z* 224.0247. For the [M + Na]^+^ ion, elimination of one and two molecules of water was observed. For cyclic oligomers having from three to seven Glc*N* units, the same losses were observed.

#### 2.7.3. Tandem Mass Spectra of Mixed Cyclooligosaccharides

The aim to reveal characteristic differences in the CID MS^2^ of protonated molecules of isomers **25** and **26** was claimed in [71]. The possibility that difference in symmetry may result in differences in their fragmentations was hypothesized. Unfortunately, CID MS^2^ of their [M + H]^+^ ions (*m*/*z* 647.2, c.e. 35 eV) contained the same signals with slight difference in relative abundance. A small peak at *m*/*z* 323 related to the disaccharide fragment [GlcGlc*N* + H]^+^ was observed for both oligosaccharides (note, its relative abundance for **25** was smaller than that for **26**, its origin for **25** is unknown).

At the same time, the differences in CID MS^2^ of the doubly charged ions [M + 2H]^2+^ (*m*/*z* 324.1, c.e. 10 eV) were much more prominent (Figure 20a,b). The disaccharide fragments for **25** were the same (*m*/*z* 324, not *m*/*z* 323, Figure 20a, inset), whereas similar cleavages for the opposite glycosidic bonds in tetrasaccharide **26** resulted in two of three possible disaccharide fragments (*m*/*z* 323 and *m*/*z* 324, Figure 20b, inset). For comparison, the [M + 2H]^2+^ ion of **21a** (Figure 20c) gave an intense peak of the same ion at *m*/*z* 323 under CID. The highest difference in CID MS^2^ of [M + 2H]^2+^ ions for compounds **25** and **26** was found in abundance of *m*/*z* 234.0971 (C_18_H_32_N_2_O_12_^2+^, [Glc*N*_2_Glc − H_2_O + 2H]^2+^, calcd. *m*/*z* 234.0972): for **25**, its relative abundance is 0.6 %, whereas for **26** it is equal to 33 %, so, in more than 50 times higher [71]. Abundances of the corresponding singly charged ion *m*/*z* 467.1869 (C_18_H_31_N_2_O_12_^+^, calcd. *m*/*z* 467.1872) were practically equal (11 % for **25** and 12 % for **26**), but the intensities of the peaks at *m*/*z* 485.1977 (C_18_H_33_N_2_O_13_) differ in an order of magnitude (5.5 % for **25** and 57 % for **26**). The data showed that eliminations of a neutral glucose residue (with the formation of a doubly charged ion) and a glucosyl cation (with the formation of a singly charged ion) are impeded for **25**, in which the glucose and glucosamine units alters by each other (it was shown in [55] for aminated β-CD that protonation occurs at amino group, so, both amino groups are protonated in [M + 2H]^2+^ of **25** and **26**), but quite possible for **26** where these residues are adjacent. It was supposed that the above processes of cyclooligosaccharide **25** should undergo through drawing together the positively charged centers in a transition state resulting in increase of energy barrier. The choice of activation was shown to be important: at c.e. 35 eV, CID MS^2^ of [M + 2H]^2+^ of **25** and **26** contained only peaks of glucosamine unit and its fragmentation products.

### 2.8. Miscellaneous

A critical review of traditional methods and recent achievements in the analysis of cyclodextrins and their derivatives is presented in [79]. This paper is devoted to comparison of strength and weakness of current official (European and US Pharmacopoeia) methods with the new advanced techniques including mass spectrometry. The aim of the review is to help analysts to decide on changing traditional analytical methods with improved new ones.

Structural description of a synthetic monosubstituted, β-CD-based glycolipid conjugates tethered through *N*-succinylamido linker was done by high-resolution positive and negative ESI MS [80]. Good agreement was obtained between experimental and calculated isotopic patterns both in positive and negative ion modes. For anions, QqTOF MS^2^ spectra were recorded; only small fragments which originated from side phospholipid chain were observed.

ESI MS was used for the characterization of regioselectively derivatized maltooligosaccharides obtained starting from natural β-cyclodextrin [81]. For two intermediates, totally acetylated and totally benzoylated heptakis-(6-deoxy-6-bromo)-β-CD, ESI HR mass spectra were reported, the only ions observed were [M + Na]^+^. For totally benzoylated derivative, QqTOF ESI MS^2^ of [M + Na]^+^ was presented and described. Successive losses of HBr (80/82 Da) and C_6_H_5_CO_2_H (122 Da) occurred. In the low-mass region, abundant ions *m*/*z* 311/313, 293/295, 189/191 corresponding to the fragmentation of a single 2,3-di-*O*-benzoyl-6-deoxy-6-bromo hexose unit along with *m*/*z* 105 (C_6_H_5_CO^+^) were observed.

An interesting observation concerning charge differentiation in tandem ESI CID mass spectra of C60:(γ-CD)_2_ inclusion complex was reported [82]. For positively charged ions [C60:(γ-CD)_2_ + 2H]^2+^ and [C60:(γ-CD)_2_ + 2Na]^2+^, only [(γ-CD) + H]^+^ or [γ-CD + Na]^+^, respectively, were registered along with ionized fragments of γ-CD ([Glc_n_ + H/Na]^+^), whereas for negatively charged ion [C60:(γ-CD)_2_ − 2H]^2−^, the major fragments gave superposed C60^−^ and [C60 + H]^−^ peak clusters along with minor superposed clusters of deprotonated cyclodextrin [γ-CD − H]^−^/[(γ-CD)_2_ − 2H]^2−^.

A mass spectrometric study of complex, supramolecular object was performed quite recently [83]. Polydisperse, α-CD-based *O*-sulfated, polyethylene glycol α,ω-dimethylacrylate-tethered, dipyrenyl-terminated rotaxanes were profiled using Orbitrap ESI MS. Positive ion mode ESI MS^2^ was successfully applied for structure characterization of individual components in a mixture of rotaxanes.

A study of unusual, separately classified compounds was published in [84]. These compounds were no real OSs, but oligomers of carbohydrate nature in which monomer units are linked by ether, not glycosidic (haemiacetal) bond. Two compounds possessing two or three glucofuranose units (eight- and eleven-membered ether cycles) in which hydroxyl groups were protected by 1,2-di-*O*-isopropylidene and 3-*O*-dodecyl, were subjected to ESI MS^2^ fragmentation in triple quadrupole instrument using Ar as a collision gas. Schemes of low-energy CID fragmentation were proposed; the main losses were acetone (from isopropylidene) and dodecanol/dodecene (from dodecyl). Ether cycle cleavage also occurred; the authors included this fission as a central part in both schemes.

## 3. Conclusions

For the last three decades, mass spectrometry of cyclic oligosaccharides has achieved great success, especially in the field of gas-phase chemistry of non-covalent, host-guest complexes. Nevertheless, some problems remain unsolved. First, there is no convenient and universal nomenclature of cleavages of cyclic oligosaccharides free of the reducing/nonreducing end formalism. No easy-to-use and reliable methods to differentiate between isomeric cyclic and acyclic oligosaccharides have been proposed. For induced decay of host-guest cyclodextrin complexes, no direct proof of guest-host proton transfer is presented. The boundaries of application of tandem mass spectrometry for determination of positions of substituents in cyclic oligosaccharides are not yet determined. A few works were done concerning fragmentation of negative ions of cyclic oligosaccharides. The methods of activation are now restricted by CID (ESI MS) and LID (MALDI MS), for example, no studies in ECD of multiply charged positive ions of cyclic CDs were carried out. A search of structure effects in complex, mixed cyclic oligosaccharides on their MS^2^ fragmentation is now in progress; we believe that discoveries will happen concerning these unusual compounds in nearest future.

## Figures and Tables

**Figure 1 molecules-24-02226-f001:**
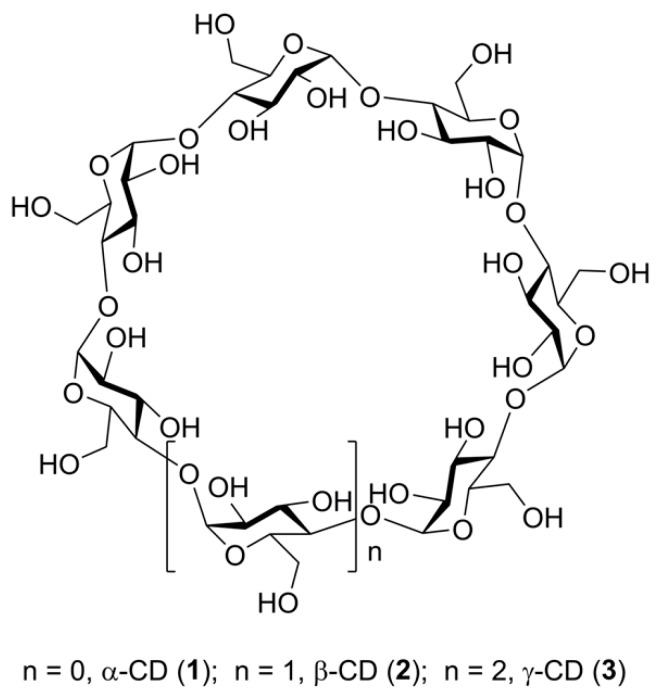
A general structural formula of α-, β-, and γ-cyclodextrins (**1**–**3**).

**Figure 2 molecules-24-02226-f002:**
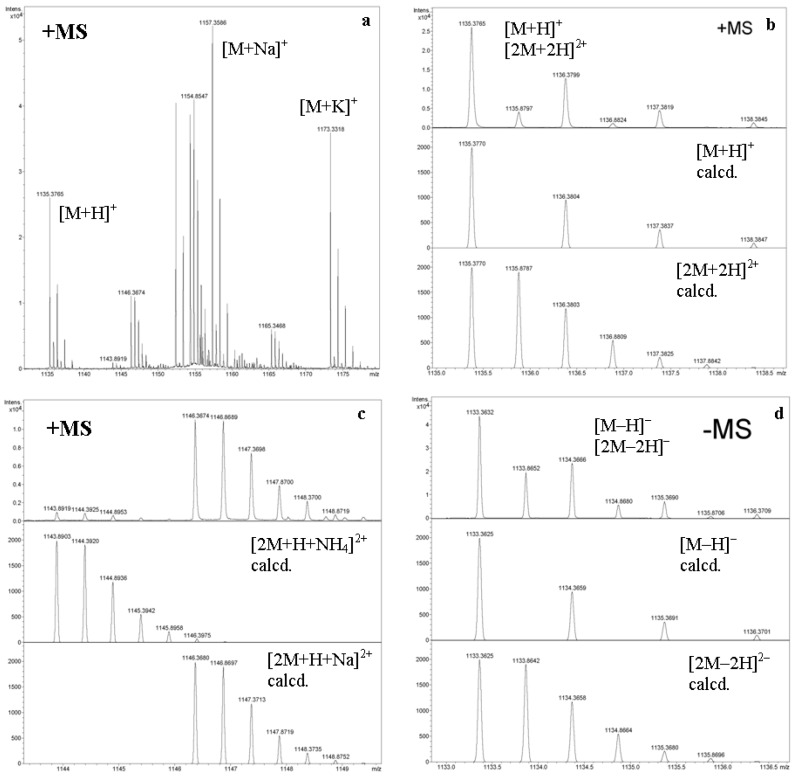
High-resolution ESI mass spectra of β-CD (the authors’ data). (**a**) Positive ion mode, range *m*/*z* 1130–1180; (**b**) same, extension *m*/*z* 1135–1138.7, (**c**) same, extension *m*/*z* 1143–1150; (**d**) Negative ion mode, range *m*/*z* 1133–1136.5. calcd.: simulated profiles of isotopic clusters.

**Figure 3 molecules-24-02226-f003:**
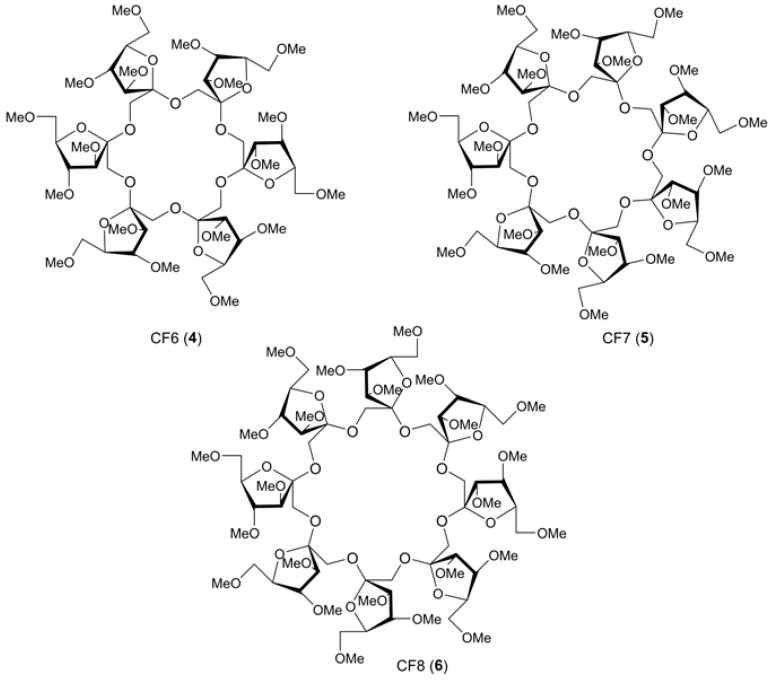
Structural formulas of cyclofructans CF6 (**4**), CF7 (**5**), and CF8 (**6**).

**Figure 4 molecules-24-02226-f004:**
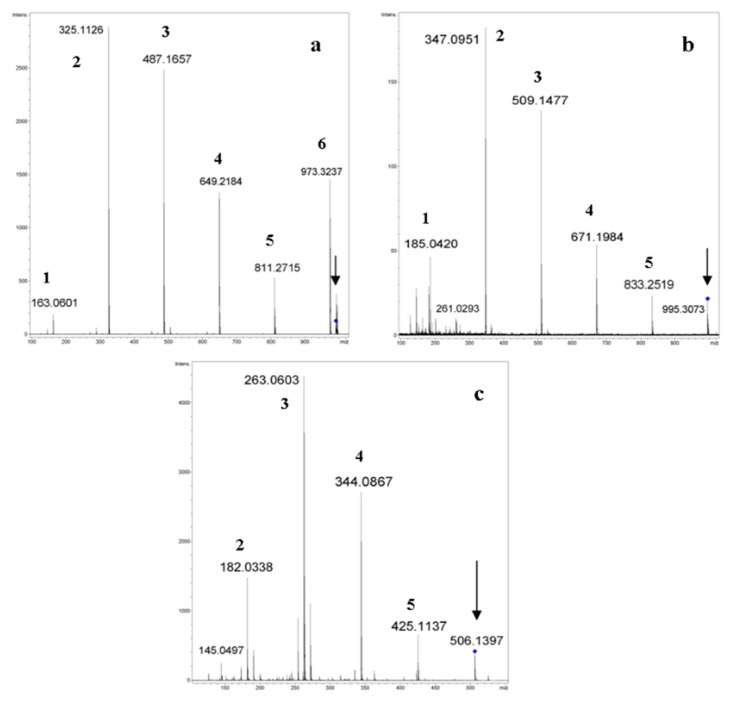
Tandem (CID MS^2^), high-resolution ESI mass spectra of selected ions generated from α-CD (the authors’ data). Fragmented ions (marked by arrows): (**a**) [M + NH_4_]^+^, *m*/*z* 990, c.e. 15 eV; (**b**) [M + Na]^+^, *m*/*z* 995, c.e. 65 eV; (**c**) [M + H + K]^2+^, *m*/*z* 506, c.e. 20 eV. The numbers mean the amount of Glc residues in the fragments [Glc_n_ + H]^+^, [Glc_n_ + Na]^+^, and [Glc_n_ + H + K]^2+^, respectively.

**Figure 5 molecules-24-02226-f005:**
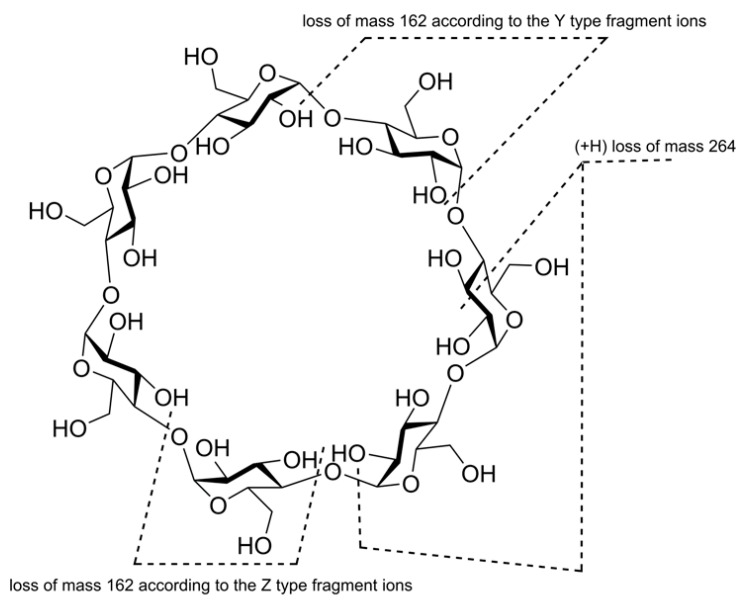
Fragmentation of [β-CD + Cat]^2+^ ions according to [33].

**Figure 6 molecules-24-02226-f006:**
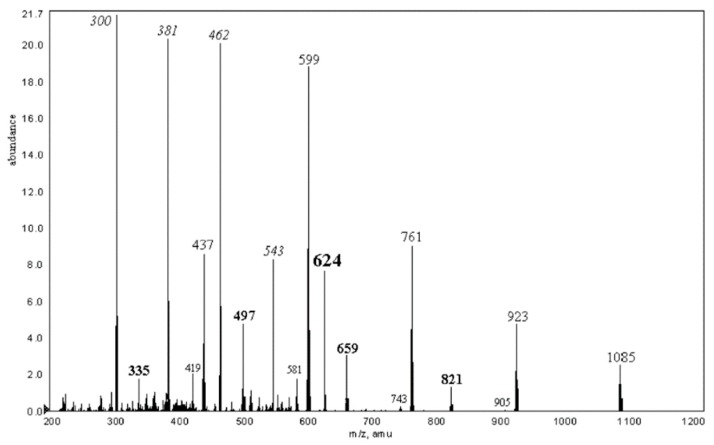
CID MS^2^ of [β-CD + Cd]^2+^ ion at *m*/*z* 624 (reproduced from [33] with permission of Elsevier).

**Figure 7 molecules-24-02226-f007:**
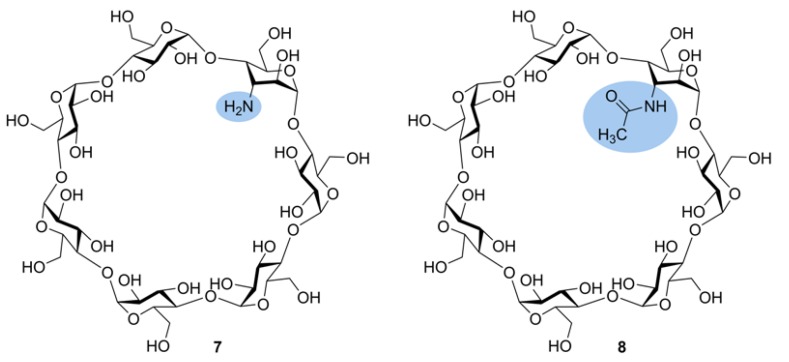
Structural formulas of monosubstituted 3-amino β-CD **7** and 3-acetamido β-CD **8** [55]. Nitrogen-containing functional groups are highlighted.

**Figure 8 molecules-24-02226-f008:**
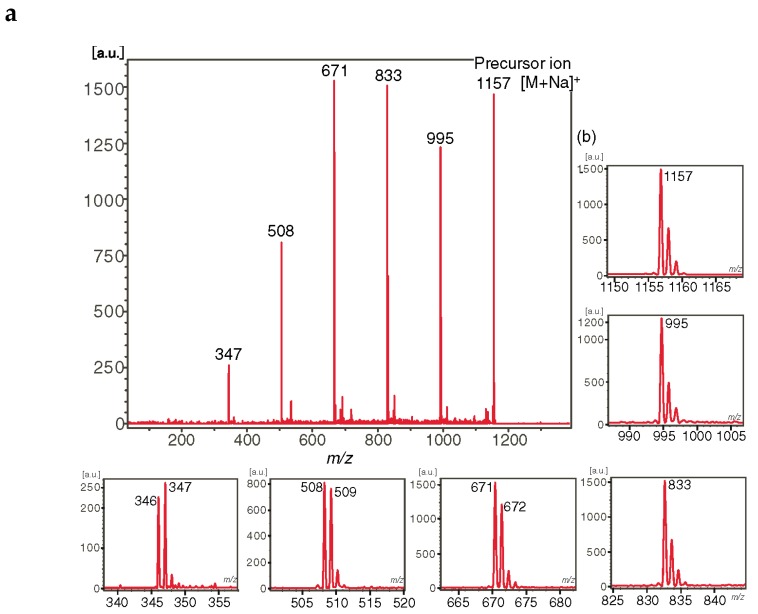
LID MALDI TOF/TOF MS of ions generated from **7**. (**a**) MS^2^ of [M + Na]^+^ ion, extensions of isotopic clusters are presented in insets; (**b**) MS^2^ of [M + H]^+^ ion (reproduced from [55], Open Access material under the Creative Commons license.).

**Figure 9 molecules-24-02226-f009:**
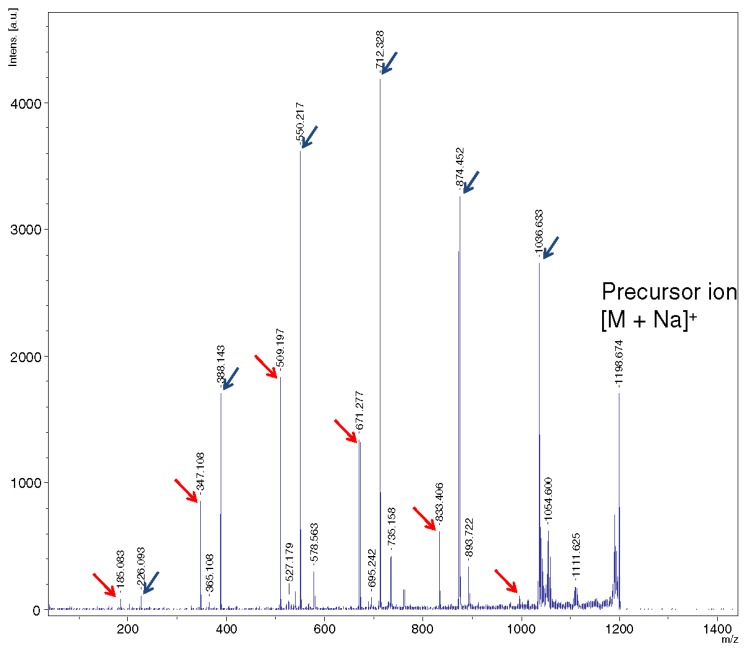
LID MALDI TOF/TOF MS of the [M + Na]^+^ ion generated from **7**. Peaks of NHAc-containing ions are marked by blue arrows, and red arrows show peaks of ions with no nitrogen (reproduced from [55], Open Access material under the Creative Commons license).

**Figure 10 molecules-24-02226-f010:**
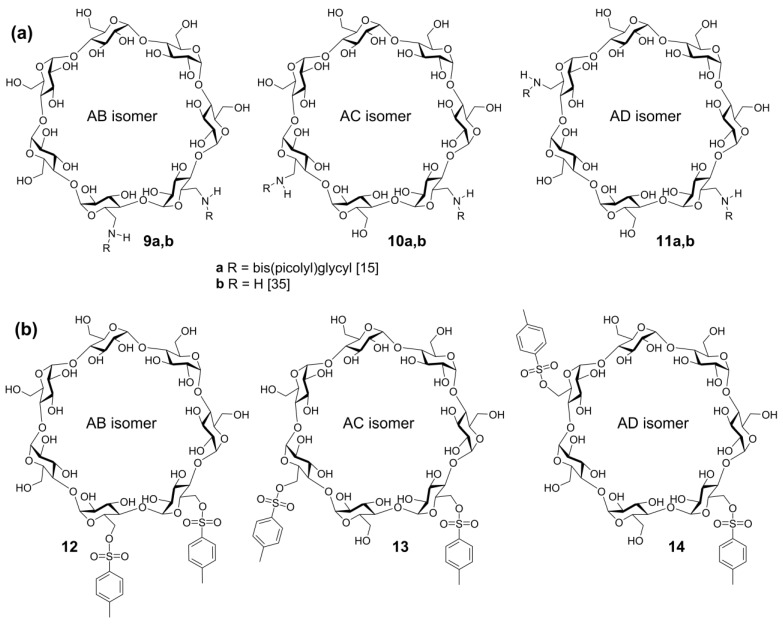
Structures of disubstituted β-cyclodextrins AB, AC, and AD (**a**) di(bis(picolyl)glycyl)amides **9a**–**11a**, diamines **9b**–**11b**; (**b**) di-*O*-tosylates **12**–**14** [15,36]). (Reproduced with changes from [36] with permission of Springer Nature).

**Figure 11 molecules-24-02226-f011:**
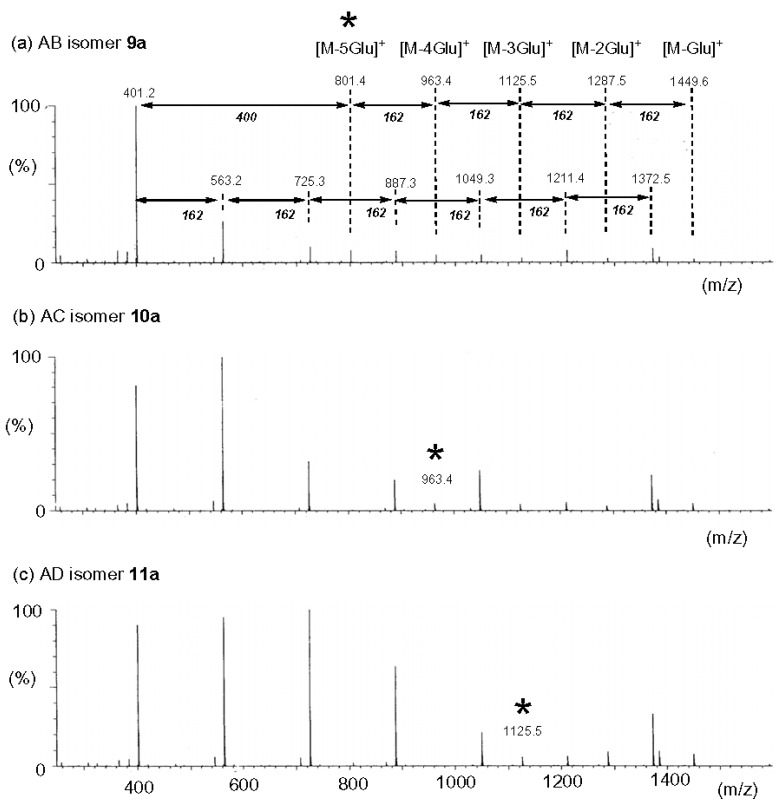
ESI CID MS^2^ of [M + H]^+^ (*m*/*z* 1612.5) of three regioisomes of di-6,6′-*N,N*′-(bis(picolyl)glycylamido)-β-CDs **9a**–**11a** (**a**–**c**). The characteristic peaks of fragment ions are marked with asterisks (reproduced from [15] with permission of SAGE Publications Ltd.).

**Figure 12 molecules-24-02226-f012:**
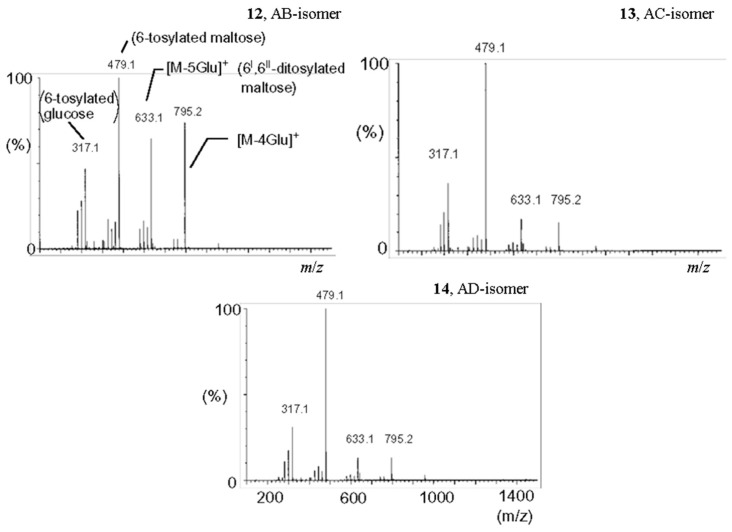
ESI CID MS^2^ of [M + H]^+^ (*m*/*z* 1443.4) of three regioisomes of di-6,6′-*O*,*O*′-tosyl-β-CDs **9a**–**11a** (reproduced from [15] with permission of SAGE Publications Ltd.).

**Figure 13 molecules-24-02226-f013:**
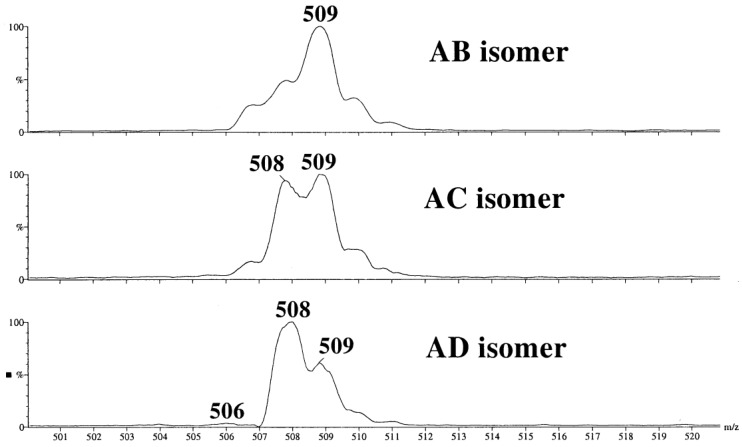
A characteristic part of pseudo-MS^2^ of [M + Na]^+^ of three regioisomeric 6,6′-diamino β-CDs (**9b**–**11b**) (reproduced from [36] with permission of Springer Nature).

**Figure 14 molecules-24-02226-f014:**
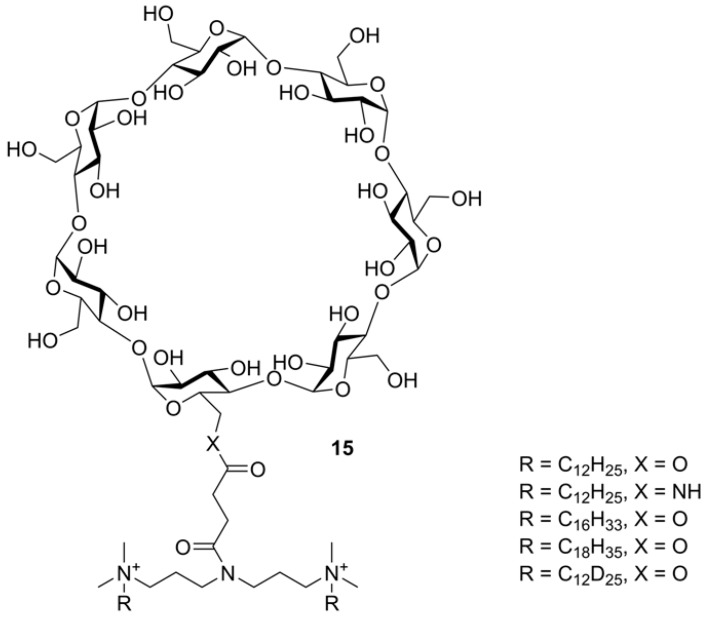
The structures of β-CD-based surfactants (reproduced from [58] with permission of Wiley).

**Figure 15 molecules-24-02226-f015:**
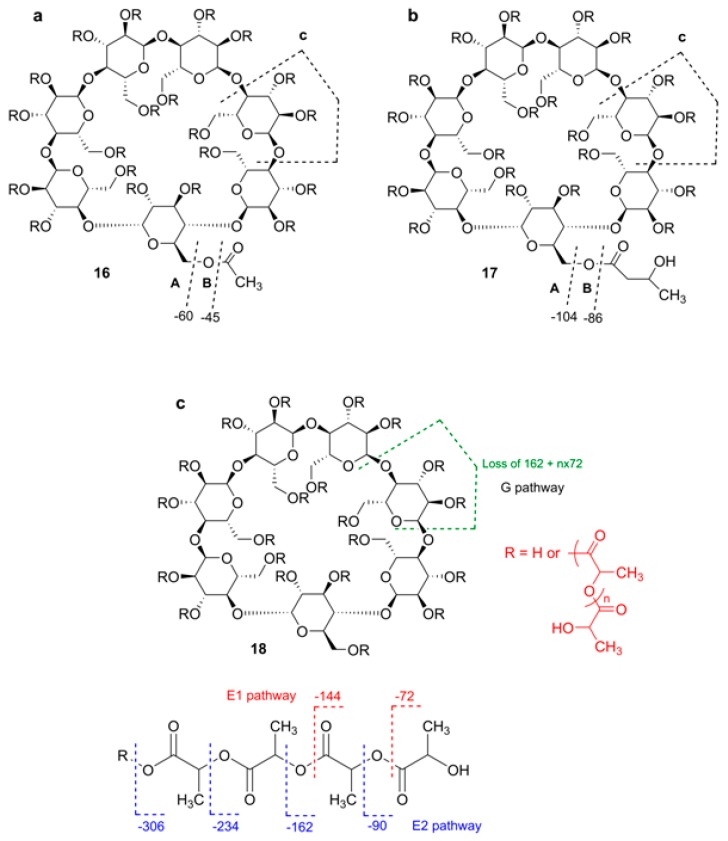
Fragmentation of acylated β-CDs according to Peptu et al. (**a**) totally acetylated, TABCD [65]; (**b**) randomly 3-hydroxybutyrated, HBCD [65]; (**c**) oligolactate [66]. (Open Access material under the Creative Commons license).

**Figure 16 molecules-24-02226-f016:**
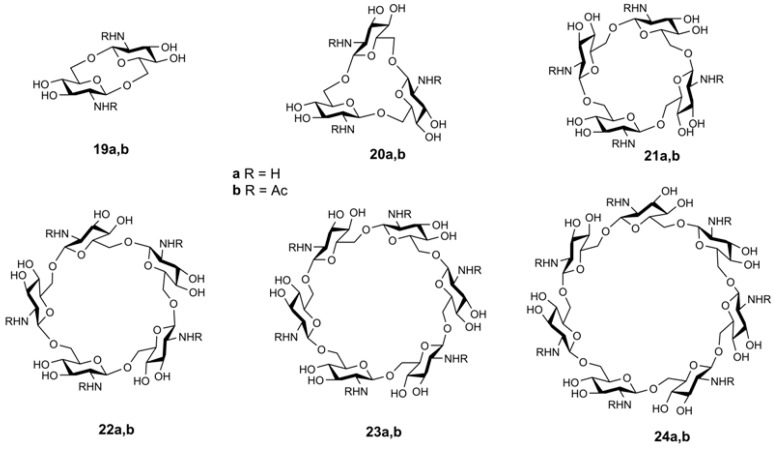
Structural formulas of cyclooligo-β-(1→6)-d-glucosamines **19a**–**24a** and cyclooligo-β-(1→6)-d-*N*-acetylglucosamines **19b**–**24b** [70,72,73,74] (reproduced from [70] with permission of Springer Nature).

**Figure 17 molecules-24-02226-f017:**
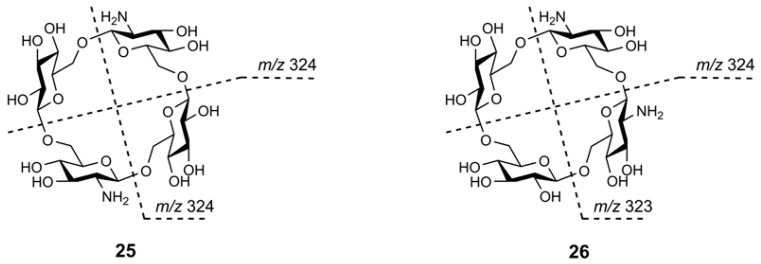
Structural formulas of cyclobis-(1→6)-(β-d-glucopyranosyl)-(1→6)-(2-amino-2-deoxy-β-d-glucopyranosyl) **25** and cyclo-(1→6)-(2-amino-2-deoxy-β-d-glucopyranosyl)-(1→6)-(2-amino-2-deoxy-β-d-glucopyranosyl)-(1→6)-(β-d-glucopyranosyl)-(1→6)-(β-d-glucopyranosyl) **26**.

**Figure 18 molecules-24-02226-f018:**
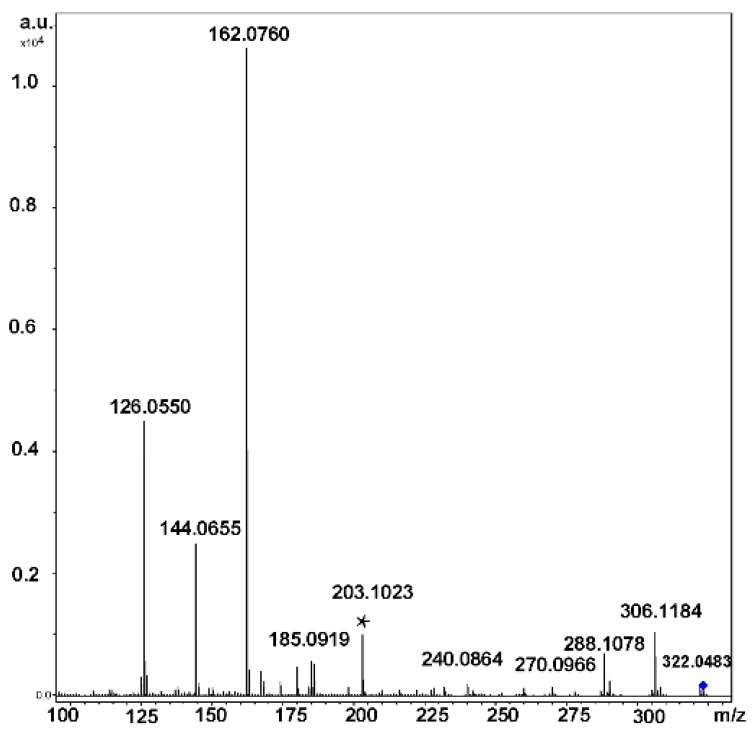
CID MS^2^ (25 eV) of the [M + H]^+^ ion of compound **19a** (reproduced from [70] with permission of Springer Nature).

**Figure 19 molecules-24-02226-f019:**
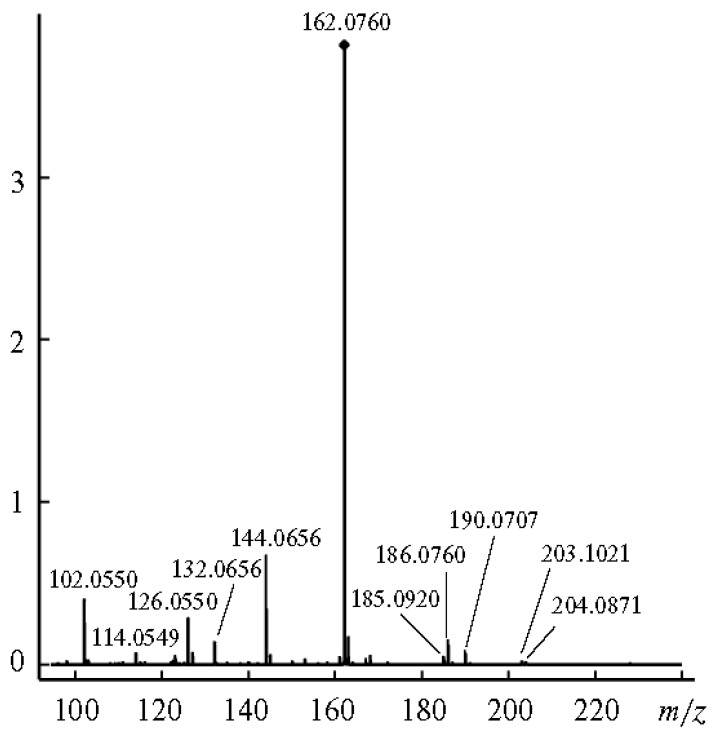
CID MS^2^ (10 eV) of the [M + 2H]^2+^ ion of compound **19a** (reproduced from [70] with permission of Springer Nature).

**Figure 20 molecules-24-02226-f020:**
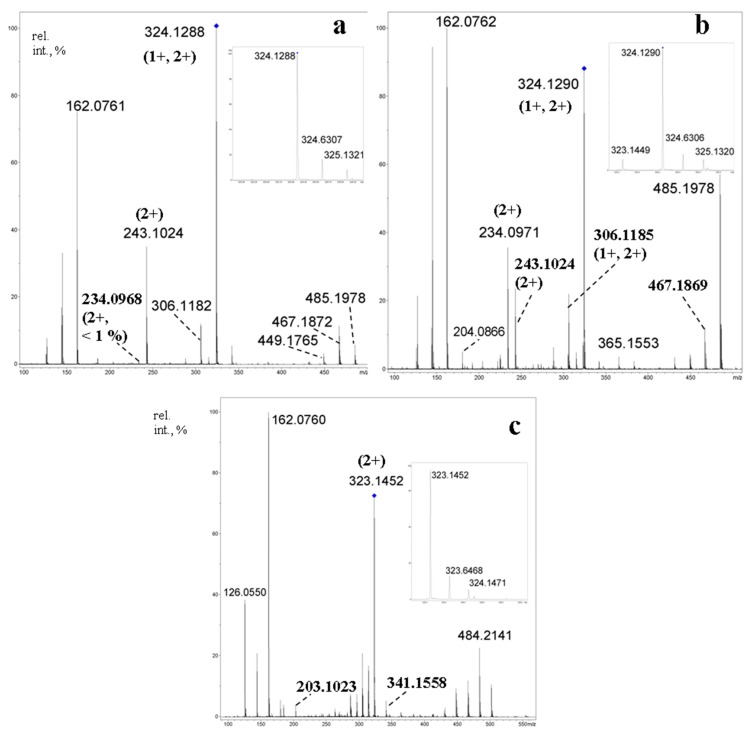
CID MS^2^ of [M + 2H]^2+^ ions for cyclotetrasaccharides **25** (**a**); **26** (**b**) and, for comparison, **21a** (**c**) (*m*/*z* 324.1 (a) and (b), *m*/*z* 323.1 (c), activation energy 10 eV). The characteristic peaks *m*/*z* 323 and *m*/*z* 324 are given in insets [71].

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
