# Peer review of "Gas-Phase Fragmentation of Cyclic Oligosaccharides in Tandem Mass Spectrometry"

_molecules, 2019, doi:10.3390/molecules24122226_

Round 1

Reviewer 1 Report

I have reviewed the manuscript, "Gas-phase Fragmentation of Cyclic Oligosaccharides in Tandem Mass Spectrometry", submitted by A. O. Chizhov and coworkers. This is a review article focuses on mass analysis of cyclic oligosaccharides. The authors discuss the current status of the topic, with extensive discussion on the MS result of various cyclic oligosaccharide derivatives, including cyclodextrins, cycloaminooligosaccharides, and cyclofructans reported in literature since 1980s. The article provides basic knowledge in this subject, which is a good starting material for beginners. However, the manuscript is immature and not in an acceptable format for following reasons.

 1. The readability of the manuscript should be improved. Using common technical terms and phrases will be helpful.  

2. The structure of the manuscript is probably acceptable, but some long sections contain too many pieces of information. I suggest the authors use subsection titles under a long section to help the reading.

3. The number of figure is insufficient. I understand that showing molecular structure of cyclic carbohydrates is important, but since the manuscript discusses spectral and fragmentation patterns, showing representative mass spectra is of equal or higher importance. Excellent examples are the review articles published by David J. Harvey (Mass Spectrometry Reviews 1999, 2006, ...). 

4. Since the overview of the fragmentation is essential in this manuscript, the important cleavage sites in the carbohydrate illustrated in figures and schemes should be indicated.

5. It will be helpful to summarize the results of comparison into tables, such as the tandem MS results in Sections 2.6 and 2.7.

6. What’s the difference between a figure and a scheme?

7. Many symbols and abbreviations are inconsistent in different paragraphs and confusing, such as those of collision energy, M (analyte or metal ion), MC2 (line 339), PQD (line 246), etc.

Author Response

Referee 1.

 1. The readability of the manuscript should be improved. Using common technical terms and phrases will be helpful.  

The comment is too general and, for this reason, it is unclear what the reviewer means. The reviewer doesn’t specify any definite phrases, paragraphs, pages etc. that have to be improved. Technical terms and phrases we use in the manuscript are generallyaccepted in mass spectrometry and carbohydrate chemistry.

2. The structure of the manuscript is probably acceptable, but some long sections contain too many pieces of information. I suggest the authors use subsection titles under a long section to help the reading.

Accepted: some topics are subdivided in the corrected version.

3. The number of figure is insufficient. I understand that showing molecular structure of cyclic carbohydrates is important, but since the manuscript discusses spectral and fragmentation patterns, showing representative mass spectra is of equal or higher importance. Excellent examples are the review articles published by David J. Harvey (Mass Spectrometry Reviews 1999, 2006, ...). 

In our opinion, general fragmentation patterns of cyclic oligosaccharides and their derivatives (figs. 3, 5, scheme 4 in the first version) are most important in a review article, nevertheless, we agree with the Reviewer 1 that some mass spectra from the original papers should be reproduced (see corrected version). In other cases, it is enough to mention characteristic ions with their m/z values in the text.

4. Since the overview of the fragmentation is essential in this manuscript, the important cleavage sites in the carbohydrate illustrated in figures and schemes should be indicated.

Figs. 3, 5 and scheme 4 (in the first version) just demonstrate the important cleavage sites in cyclic oligosaccharides.

5. It will be helpful to summarize the results of comparison into tables, such as the tandem MS results in Sections 2.6 and 2.7.

The comment is unclear: what definitely should be tabulated? In the corrected version, we present the spectra from the original publications, the principal peaks are marked.

6. Whats the difference between a figure and a scheme?

Corrected. Now there are only figures in the manuscript.

7. Many symbols and abbreviations are inconsistent in different paragraphs and confusing, such as those of collision energy, M (analyte or metal ion), MC2 (line 339), PQD (line 246), etc.

Some abbreviations have been presented more definitely and added to the list of abbreviations.

Accepted: M is a molecule of analyte, an abbreviation of "metal" (associated cation) is Cat.

Accepted: MC2 (line 343) is a misprint, should be MS2.

PQD (line 246).

Added to the list of abbreviations: PQD, pulsed-Q-dissociation.

Reviewer 2 Report

This review article tries to “discuss” about the mass spectral analyses of cyclodextrin-related compounds. A reviewer think this focus is interesting knowing that it is very difficult compared to the usual glycan analysis. However, a reviewer feels somewhat uncomfortable.  Please see my major concerns.

Major concerns

This review tended to point out errors and problems of others published data.  Although it may be true, this is not common as far as this reviewer knows. This reviewer strongly suggests reconsidering the way of expressions in line 55, 92, 135, 180, 252, 283, 353, and 361.

In the criticism in around line 135, this reviewer got their point. However, Y-type ion formation mechanism is same as that of B-ion formation. So that the type of product ions may better be expressed as B/Y-ion. This expression is used to describe products produced for OS where both reducing and non-reducing carbohydrates were cleaved leaving a middle structure.

A chapter 2.7 sounded just like a part of a regular article without MS spectra. Up until this section, a reviewer thought that the review tries to summarize a way of discriminating the CD derivatives, but this chapter explains details of individual fragmentation process only. An array of synthesized samples are truly unique and so readers may want to know the difference in fragmentation process. In line 541, unexpected observation of m/z 323 was described without any explanation.

Minor and/or technical points

Many abbreviations were used and it was shown in the end of the manuscript. It is much easier if it is presented at the beginning.

line 53
It may be okay but, this reviewer have never heard of “second-order MS”, which is defined as “tandem MS” later in the Fig. 2 legend. Define when it is used first.

line 141 & 142
It is not appropriate to indicate the page number of a reference.

line 276
What is the path B?

line 369
CD nucleus may mean CD core.

Unity issue
This reviewer did not understand the differences in Schemes and Figures.
Different numbering of compounds were used thus hard to follow.
This reviewer tends to understand but the presentation of m/z values with different places of decimals looks odd.
CE, c.e, and c.e. were used to describe probably collision energy.
C60 v.s C60

This reviewer think “link” is not right word. A “unit” may be the word, but the word was also used made me wonder. What is the difference between them. Readers may not understand.

MS signals are not usually called as peaks.

The Greek letters are missing in the reference titles.

Author Response

Referee 2.

Major concerns

This review tended to point out errors and problems of others published data.  Although it may be true, this is not common as far as this reviewer knows. This reviewer strongly suggests reconsidering the way of expressions in line 55, 92, 135, 180, 252, 283, 353, and 361.

In our opinion, a review article should not be a simple compilation of published data but has to include also a critical analysis of those data and revealing existing problems in the field (lines 92-99, 135-156). We find it important to mention such problems as their solution may contribute to further development of MS of cyclic OSs. A search for arguable conclusions (lines 181, 254-256, 284-287, 351-353) or mistakes (lines 54-57) in cited publications was by no means the goal of the review, however, the authors of the review have the right to evaluate published data and express their disagreement or doubt if some conclusions seem to be insufficiently substantiated.

In the criticism in around line 135, this reviewer got their point. However, Y-type ion formation mechanism is same as that of B-ion formation. So that the type of product ions may better be expressed as B/Y-ion. This expression is used to describe products produced for OS where both reducing and non-reducing carbohydrates were cleaved leaving a middle structure.

The designation of "B/Y-ion" may be possibly applicable for the easiest case of homocyclooligosaccharides, for example, intact cyclodextrins. If one or more substituents or different carbohydrate residue(s) is (are) introduced, this description would be inappropriate, as well as "A/X-ions" (in other words: how to evaluate subscribes and superscribes?).

A chapter 2.7 sounded just like a part of a regular article without MS spectra.

Mass spectra are added in the corrected version

Up until this section, a reviewer thought that the review tries to summarize a way of discriminating the CD derivatives, but this chapter explains details of individual fragmentation process only. An array of synthesized samples are truly unique and so readers may want to know the difference in fragmentation process. In line 541, unexpected observation of m/z 323 was described without any explanation.

This comment is not clear. The origin of the ion with m/z 323 is explained in scheme 5 (the first version).

Minor and/or technical points

Many abbreviations were used and it was shown in the end of the manuscript. It is much easier if it is presented at the beginning.

The position of the list of abbreviation is in agreement with guidelines for authors.

line 53
It may be okay but, this reviewer have never heard of “second-order MS”, which is defined as “tandem MS” later in the Fig. 2 legend. Define
when it is used first.

Technical terms and phrases we use in the manuscript are generally accepted in mass spectrometry. (K.K. Murray, R.K. Boyd, M.N. Eberlin, G.J. Langley, L. Li, Y. Naito, Definitions of terms relating to mass spectrometry (IUPAC Recommendations 2013) // Pure Appl. Chem. 2013. Vol. 85, N 7. P. 1515–1609).

line 141 & 142
It is not appropriate to indicate the page number of a reference.

This is necessary for description of the cited article.

line 276
What is the path B?

This is the designation given by the authors of the previously cited paper [50].

line 369
CD nucleus may mean CD core.

Accepted, "core".

Unity issue
This reviewer did not understand the differences in Schemes and Figures.

Accepted, all are "Figures"

Different numbering of compounds were used thus hard to follow.

Accepted: unified numbering through the review is introduced in the corrected version.

This reviewer tends to understand but the presentation of m/z values with different places of decimals looks odd.

Decimals are presented in HRMS.

CE, c.e, and c.e. were used to describe probably collision energy.

List of Abbreviations: ...CE, capillary electrophoresis, c.e., collision energy...

C60 v.s C60

Accepted: C60

This reviewer think “link” is not right word. A “unit” may be the word, but the word was also used made me wonder. What is the difference between them. Readers may not understand.

(Carbohydrate) "unit", "residue", "link" are synonyms in the context, nothing wonderful. Technical terms and phrases we use in the manuscript are generally accepted in mass spectrometry and carbohydrate chemistry.

MS signals are not usually called as peaks.

MS signals are usually named as "peaks", see, for example, references [8, 9, 10, 15, 17, 35 etc.]

The Greek letters are missing in the reference titles.

A bug of transfer. We'll try to correct it manually on the site.

Round 2

Reviewer 1 Report

The authors have addressed most of my concerns except the first comment. Here I state my first comment more precisely.

1. The manuscript is not written with the standard English. So,  many descriptions cause confusion and misunderstanding. Just a few simple examples:

(a) Line 35-38, “Soft ionization methods (primarily, ESI and MALDI) make possible to transfer heavy, polar molecules into gas phase provide complementary, mass spectrometric approach [8] for studies of macrocycles (including CDs) along with NMR, UV/Vis spectroscopy, circular dichroism, chromatography, etc. [4].”, the meaning of this sentence is confusing to me.

(b) Line 47-50, “For cyclodextrins (a-CD, cyclohexamaltose, 1; b-CD, cycloheptamaltose, 2; g-CD, cyclooctamaltose, 3, Fig. 1), FAB MS [9], and then, ESI (named there "ion evaporation atmospheric-pressure ionization mass spectrometry") [11] has been reported.”, I think the authors want to say something like “The mass spectra of cyclodextrins obtained using FAB and ESI MS have been reported (in 2004 and 1990, respectively).”. By the way, there is no description in the main text referring to Ref 10, I suppose the authors mean “FAB MS [9,10]”.

I cannot list all language issues in the entire manuscript. Improving the readability is important not only for native English readers, but also for non-native English readers who rely on more precise descriptions.

2. Common technical terms are recommended. For example:

a.      Tandem MS is more appropriate to second-order MS.

b.      FT-ICR MS is more appropriate to ICR FT MS.

c.       Permethylation is more common than totally methylation.

        There are some other minor problems in the technical terms but they are probably acceptable.

Author Response

Reviewer 1-2

The authors have addressed most of my concerns except the first comment. Here I state my first comment more precisely.

1. The manuscript is not written with the standard English. So,  many descriptions cause confusion and misunderstanding. Just a few simple examples:

(a) Line 35-38, “Soft ionization methods (primarily, ESI and MALDI) make possible to transfer heavy, polar molecules into gas phase provide complementary, mass spectrometric approach [8] for studies of macrocycles (including CDs) along with NMR, UV/Vis spectroscopy, circular dichroism, chromatography, etc. [4].”, the meaning of this sentence is confusing to me.

Accepted: the phrase is changed (divided in two).

Soft ionization methods (primarily, ESI and MALDI) make possible to transfer heavy, polar molecules into gas phase. This achievement opened a way for complementary, mass spectrometric approach [8] for gas-phase studies of macrocycles (including CDs) in addition to NMR, UV/Vis spectroscopy, circular dichroism, chromatography, etc. used for liquid phase studies [4].

(b) Line 47-50, “For cyclodextrins (a-CD, cyclohexamaltose, 1; b-CD, cycloheptamaltose, 2; g-CD, cyclooctamaltose, 3, Fig. 1), FAB MS [9], and then, ESI (named there "ion evaporation atmospheric-pressure ionization mass spectrometry") [11] has been reported.”, I think the authors want to say something like “The mass spectra of cyclodextrins obtained using FAB and ESI MS have been reported (in 2004 and 1990, respectively).”

Accepted in part: description of CDs is transferred into beginning of Introduction.

For cyclodextrins, FAB MS [10], and then, ESI (named there as "ion evaporation atmospheric-pressure ionization mass spectrometry") [11] has been reported. A year later, an extensive study of electrospray (named as "ion spray", IS) MS has been done for intact and partially alkylated, partially acylated a-, b-, and g-CDs using pure solvents and inorganic dopants (salts) in a positive ion mode [12].

. By the way, there is no description in the main text referring to Ref 10, I suppose the authors mean “FAB MS [9,10]”.

No, the ref. is misprinted ([10] should be instead of [9])

I cannot list all language issues in the entire manuscript. Improving the readability is important not only for native English readers, but also for non-native English readers who rely on more precise descriptions.

2. Common technical terms are recommended. For example:

a.      Tandem MS is more appropriate to second-order MS.

That is not true: "tandem MS" is a general term for second-order MS (MS2), third-order MS (MS3), fourth-order MS (MS4), etc. Of course, if "tandem" is mentioned, one can usually presume it as MS2.

b.      FT-ICR MS is more appropriate to ICR FT MS.

Accepted.

c.       Permethylation is more common than totally methylation.

Accepted in part (where "totally" is not included in abbreviation, as in TABCD or "totally or partially").

        There are some other minor problems in the technical terms but they are probably acceptable.

Reviewer 2 Report

All the points raised have  been addressed in a way of the authors. Although a reviewer do not entirely agree with authors' rebuts, this is up to the authors considering this is a review article. A reviewer is not happy about a term "second-order MS", which is not included in the Definitions of terms relating to mass spectrometry (IUPAC Recommendations 2013) despite the authors' claim. A reviewer assume it is used in some group. One point left behind is one of the unity issue. CE is used as the abbreviation of capillary electrophoresis and collision energy as well (line 256). Regarding the "carbohydrate link", a reviewer change the way of pointing out. The term is NOT used in the carbohydrate field at least for couple of decade as far as a reviewer knows. It is not just odd but also confusing because some uses the term to describe protein-carbohydrate interaction, which is not common also. It is better replacing link with other term like unit.

Author Response

Reviewer 2-2

All the points raised have  been addressed in a way of the authors. Although a reviewer do not entirely agree with authors' rebuts, this is up to the authors considering this is a review article. A reviewer is not happy about a term "second-order MS", which is not included in the Definitions of terms relating to mass spectrometry (IUPAC Recommendations 2013) despite the authors' claim. A reviewer assumes it is used in some groups. One point left behind is one of the unity issue. CE is used as the abbreviation of capillary electrophoresis and collision energy as well (line 256).

Nevertheless, the abbreviation "MSn" is commonly used in mass spectrometry and related fields. Of course, "n" is a number of steps of activation (n = 1 means generation of ions which are inevitably activated in comparison to the initial molecule). Like in mathematics "n" means an order of magnitude, in tandem mass spectrometry it means a step (or order) of activation. So, "n-order mass spectrum" is a convenient verbal description of the designation MSn.

CE: the term CE50(collision energy needed for decay of 50 % of the fragmented ion), is transferred from the cited paper and it is commonly accepted. CE50is added to the list of abbreviations.

Regarding the "carbohydrate link", a reviewer change the way of pointing out. The term is NOT used in the carbohydrate field at least for couple of decade as far as a reviewer knows. It is not just odd but also confusing because some uses the term to describe protein-carbohydrate interaction, which is not common also. It is better replacing link with other term like unit.

Accepted: the term "link" is replaced by "residue" and "unit" where it is possible. As for "carbohydrate link", the derivatives "linkage" and "linked" are widely used.
